# Master transcription-factor binding sites constitute the core of early replication control elements

Jesse L Turner [ID] [1,2,7], Laura Hinojosa-Gonzalez [ID] [3,4,7], Takayo Sasaki [ID] [2], Satoshi Uchino [ID] [2], Athanasios Vouzas[1,2], Mariella S Soto [ID] [2], Abhijit Chakraborty[3], Karen E Alexander[5], Cheryl A Fitch[1], Amber N Brown [ID] [1], Ferhat Ay [ID] [3,4,6 ✉] & David M Gilbert [ID] [2 ✉]

## Abstract

**Eukaryotic genomes replicate in a defined temporal order called the replication timing (RT) program. RT is developmentally regulated with the potential to drive cell fate transitions, but mechanisms controlling RT remain elusive. We previously identified "Early Replication Control Elements" (ERCEs), cis-acting elements necessary for early RT, domain-wide transcription, 3D chromatin architecture and compartmentalization in mouse embryonic stem cells (mESCs), but deletions identifying ERCEs were large and encompassed many putative regulatory elements. Here, we show that ERCEs are compound elements, whose RT activity can largely be accounted for by multiple binding sites for diverse master transcription factors (subERCEs). While deletion of subERCEs had large effects on both transcription and replication timing, deleting transcription start sites eliminated nearly all transcription with only moderate effects on replication timing. Our results suggest a model in which subERCEs are a class of transcriptional enhancers that can also organize chromatin domains structurally to support early replication timing, potentially providing a feed-forward loop to drive robust epigenomic change during cell fate transitions.**

**Keywords** Cell Cycle; Replication Timing; Transcription; Genome Architecture; Cell fate Transitions
**Subject Categories** Chromatin, Transcription & Genomics; DNA Replication, Recombination & Repair

## Introduction

DNA replication is an essential process that facilitates the faithful duplication of genetic information. In eukaryotes, chromosomes are replicated in a defined temporal order called the replication timing (RT) program. In mammals, changes in RT occur during development in units of 400-800 kilobases termed replication domains (RD), and changes in RT during development correlate with changes in transcriptional programs (Rivera-Mulia et al, 2019a; Vouzas and Gilbert, 2023). Defects in RT are correlated with dysregulation of genes and genome instability, observed in several diseases and cancers, and associated with defects in chromosome condensation and sister chromatid cohesion (Blumenfeld et al, 2017; Heskett et al, 2020; Platt et al, 2018; Rivera-Mulia et al, 2017; Rivera-Mulia et al, 2019b; Sasaki et al, 2017). Recently, it was revealed that RT perturbations result in epigenetic disturbances (Klein et al, 2021), demonstrating that RT is essential to maintain the epigenome.

We previously demonstrated (Sima et al, 2019) that early RT of several RDs in mouse embryonic stem cells (mESCs) is controlled by discrete, cooperative, and redundant chromosome segments termed early replication control elements (ERCEs). ERCEs are required for early RT, transcription, euchromatic compartmentalization, and Topologically Associating Domain (TAD) organization of their resident domain. However, the deletions identifying ERCEs spanned tens of kilobases and multiple discrete regions with active chromatin signatures, leaving open the possibility that they may contain several different types of regulatory elements. ERCEs contain enhancer histone marks, the histone acetyltransferase CBP/P300, multiple Oct4/Sox2/Nanog co-bound (OSN) sites, transcription start sites (TSS), and they interact in a CTCF-independent manner. Changes in cooperative binding of master transcriptional regulatory network (TRN) transcription factors (TFs) have been shown to strongly correlate with changes in RT during differentiation (Rivera-Mulia et al, 2019a). In budding yeast, TFs Fkh1/Fkh2 mediate early RT through spatial clustering of early origins, independent of their transcriptional activity (Fang et al, 2017; Knott et al, 2012; Ostrow et al, 2017). Together, these results suggest the hypothesis that ERCE activity may reside within their resident master TF co-binding sites.

To test this hypothesis, we performed a series of finer deletions to delineate the locations of ERCE activity. We found that OSN sites contributed to RT activity in all three regions, in some contexts completely accounting for ERCE activity. By contrast, deletion of all three TSSs that resided within the prior deletions eliminated nearly all transcription in the domain without loss of

[1]Department of Biological Science, Florida State University, Tallahassee, FL 32306, USA. [2]San Diego Biomedical Research Institute, San Diego, CA 92121, USA. [3]La Jolla Institute for Immunology, La Jolla, CA 92037, USA. [4]Bioinformatics and Systems Biology PhD Program, University of California, San Diego, CA, USA. [5]Department of Biomedical Sciences, College of Medicine, Florida State University, Tallahassee, FL 32306, USA. [6]Department of Pediatrics, University of California San Diego, La Jolla, CA 92093, USA. [7]These authors contributed equally: Jesse L Turner, Laura Hinojosa-Gonzalez. ✉E-mail: ferhatay@lji.org; gilbert@sdbri.org

early replication. Thus, early replication can be maintained in the face of severe reductions in transcriptional activity. We find that OSN sites are also bound by many diverse master TFs and are unlikely to derive their RT activity from the OSN TFs themselves designate these sites as "subERCEs", novel functional elements that can enhance transcription but also play nontranscriptional roles in RT. subERCEs do not overlap efficient replication origins, and a large deletion encompassing most of the replication initiation activity in the domain did not affect RT, demonstrating that subERCEs regulate RT domain-wide, independent of where replication initiates. Together with our prior demonstration of their 3D interactions, supported by more recent HiChIP and Micro-C data (Jusuf et al, 2025; Kraft et al, 2022; Reyna et al, 2024; Cattoglio et al, 2019; Hansen et al, 2017), we propose that ERCEs are compound elements that create a subnuclear environment that is favorable for early firing of potential replication origins located throughout the domain. Notably, our dissections highlight the robustness and cooperativity of ERCEs, which function as redundant compound cis-regulatory elements to maintain early replication.

# Results

## Functional dissection of the Dppa2/4 replication domain using allele-specific deletions

The ~400 kb murine "Dppa domain" contains three active genes in mESCs, *Dppa2*, *Dppa4*, and *Morc1* (Fig. EV1A). *Dppa2* and *Dppa4* are markers of pluripotency (Maldonado-Saldivia et al, 2007) and *Morc1* is essential for spermatogenesis (Inoue et al, 1999), although it is also highly expressed in ESCs (Sima et al, 2019). The Dppa domain corresponds to a TAD whose boundaries align with Hi-C A/B compartment boundaries and are demarcated by convergent CTCF binding sites that are also bound by cohesin (Sima et al, 2019). Our previous study identified three segments with ERCE activity in the domain, designated A, B, and C (Fig. EV1A,B). These ERCE-containing segments interact and contribute to TAD strength in a CTCF-independent manner (Sima et al, 2019). ERCEs, but not CTCF protein or the CTCF-bound TAD boundaries, were necessary to maintain interactions with the A compartment and early replication. The Dppa domain resides in the nuclear interior but is flanked on both sides by late-replicating lamina-associated domains (Takebayashi et al, 2012a), and deletion of the ERCE-containing segments causes a movement of the Dppa domain toward the nuclear lamina (Brueckner et al, 2020).

During differentiation, the Dppa domain undergoes a domain-wide shift from early to late RT (Fig. EV1Bi), concomitant with a switch from the A compartment to the B compartment and silencing of genes throughout the domain (Hiratani et al, 2010; Hiratani et al, 2008; Sima et al, 2019; Takebayashi et al, 2012b). These changes occur in all three germ layers, thus the Dppa domain is a pluripotency-specific early replicating domain (Hiratani et al, 2010; Hiratani et al, 2008).

Our studies of *cis*-acting elements in the Dppa domain have been greatly facilitated by the use of mESCs derived from F1 *castaneusXmusculus* hybrid mice (F121-9). These two subspecies are separated by 500,000 years and harbor single-nucleotide polymorphisms (SNPs) at, on average, every 150 bp, facilitating

distinction between paternal and maternal homologs (Rivera-Mulia et al, 2018). This has allowed us to make heterozygous manipulations in one allele (red lines in all figures) while retaining the WT allele in the same datasets (black lines in all figures) (Fig. EV1C). Figure EV1Bii–viii shows the results of single, double, and triple ERCE deletions that were previously published (Sima et al, 2019).

Given that we expected a series of smaller deletions to have more subtle effects on RT than our original deletion series, we took advantage of our hybrid system to develop a rigorous statistical method to determine the RT change for each deletion with respect to its corresponding internal control (WT) allele. This approach compares the area between two RT curves of the Dppa domain (WT vs deletion; Fig. EV1C) to the area from similarly-sized random regions across the genome to test whether the observed delay for the Dppa domain is significantly larger than expected by chance (Figs. EV1D and EV2A,B; see "Methods"). Upon testing this approach on our previously published deletions, we observed statistically significant differences for ΔABC and all pairwise deletions as expected (Fig. EV1D). Among the single deletions, statistical significance was reached by B and C deletions but not for A deletion, which is consistent with visual patterns of RT curves (Fig. EV1D). These results demonstrate the effectiveness of our new quantitative approach, which becomes critical to assess the impact of smaller deletions while incorporating data from multiple deletion clones or replicates (Fig. EV2; see "Methods"). These numbers are provided within each deletion panel in all figures throughout the manuscript (blue font; Δarea and pval in each RT panel). Figure EV3 shows all of the individual replicates from independent CRISPR-mediated deletion clones we generated for this study or re-analyzed from Sima et al. Figure EV2B compares the |Δarea| and pval for all of these same deletions. Tables EV1, 2, and 3 contain the sgRNA/primer sequences, coordinates of deletions as determined by PCR/Sanger sequencing/NGS analysis, and the genotypes of 65 cell lines, respectively.

## Cooperative contributions of 3 OSN sites maintain the activity of ERCE C

We first performed a series of deletions in the 39 kb ERCE C, the ERCE deletion with the largest effect on RT when deleted alone (Fig. EV1B). ERCE C contains three OSN sites (COSN3, COSN2, COSN1, 5′ to 3′) and one annotated intragenic TSS for Morc1 ~135 kb downstream from the start of the gene (CTSS) (Fig. 1A,B). To ensure that our classification of OSN sites was supported by multiple independent ChIP experiments, and to provide a quantitative assessment of the strength of each OSN site, we interrogated the Remap2022 database for mESC datasets (Hammal et al, 2022) and quantified the percentage of the available datasets for each TF that contained a peak overlapping our targeted deletions. This analysis revealed that OSN binding activity is detected at CTSS in some available datasets but much less reproducibly than the sites we refer to in this manuscript as OSN sites (Fig. 1B). Since the right half of C contains all active regulatory marks (Figs. 1A and EV1A) and an annotated mESC super-enhancer (Dowen et al, 2014; Khan and Zhang, 2016), we began by deleting the right and left halves of C. Deletion of the left half (ΔCL20k) had no effect while deletion of the right half (ΔCR14.6k) caused a significant delay that was reproducible in two independent deletion clones (Figs. 1Ci,ii,D and EV3xxvi,xxxii). We next focused

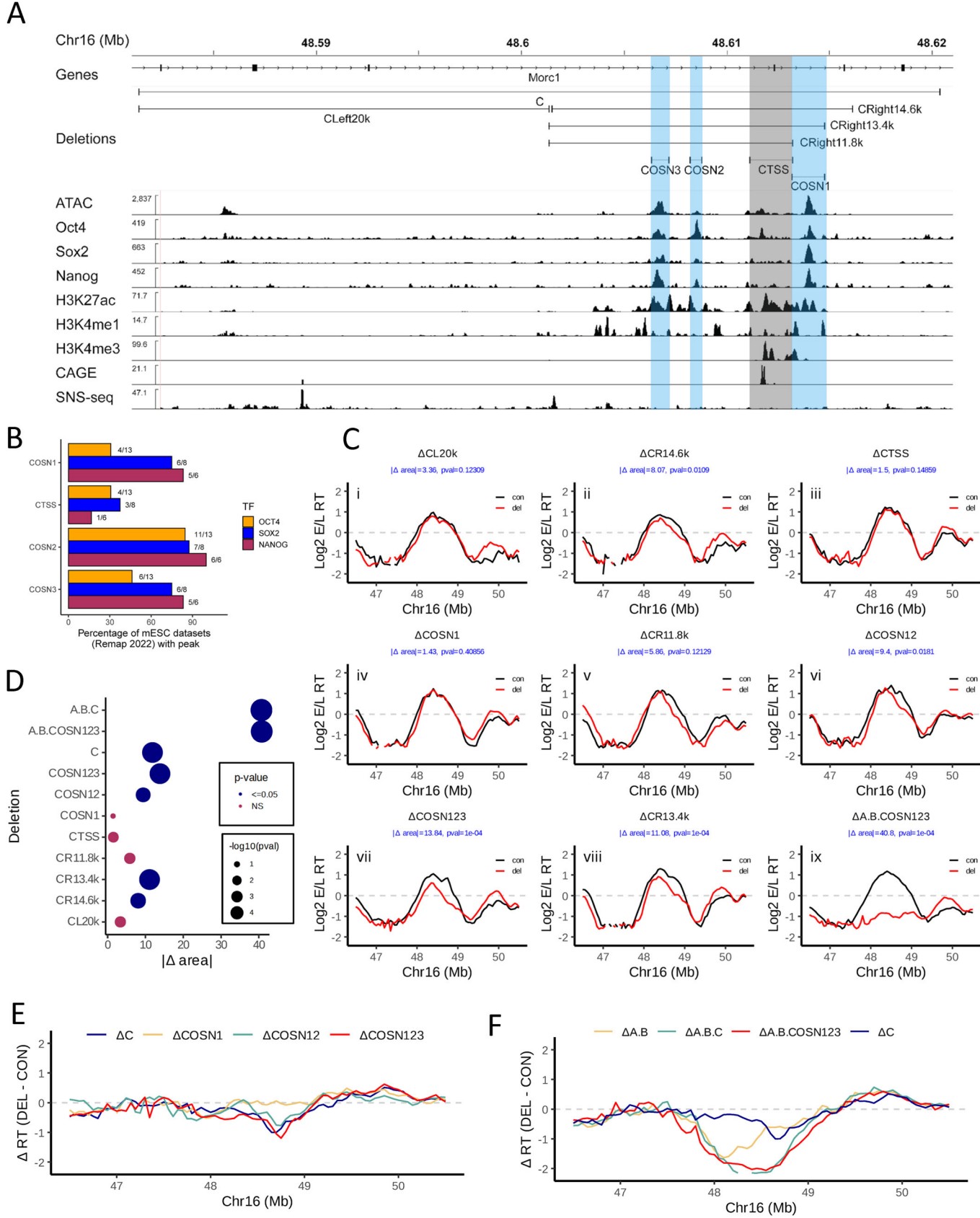

**Figure 1. Genetic dissection of the C element implicates OSN binding regions as central to ERCE RT activity.**

(A) IGV browser tracks of the Dppa domain, zooming into the C element. Positions of OSN and TSS are indicated in blue and gray, respectively. All deletions of the C element are represented by horizontal bars above the data tracks. (B) Percentage of Remap2022 mESC Oct4, Sox2, and Nanog ChIP-seq datasets with peaks that overlap the deletions generated in the C element. (C) RT profiles in the Dppa domain of cell lines harboring allele-specific deletions (red) compared to the homologous unmodified allele (black) in (C) that are shown in (A), averaged across replicates (independent CRISPR clones and technical replicates). Individual replicate experiments are shown in Fig. EV3. Δarea and pval, calculated as in Fig. EV2, are shown in blue font in each RT panel. (D) |Δ area| between the RT profile curves for the deletions shown in (C) and the significance of their delay in RT compared to the corresponding WT alleles (see "Methods"). Empirical P values are shown. (E, F) Comparison of RT changes [ΔRT (DEL-CON)] in the Dppa domain (E) comparing ΔC to sequential deletions of COSN elements (Sima et al, 2019) and COSN123, and (F) comparing ΔA.B.C (Sima et al, 2019) to ΔA.B.COSN123 with ΔAB and ΔC as comparators. Source data are available online for this figure.

our attention on the OSN and TSS sites in the right half of C. We deleted the TSS (ΔCTSS; Fig 1Ciii), the OSN site downstream of the TSS (ΔCOSN1; Fig. 1Civ), as well as a deletion of COSN2, CTSS and COSN3 that leaves COSN1 intact (ΔCR11.8k; Figs. 1Cv and EV3xxx), none of which resulted in a statistically significant change in RT compared to WT (Fig. 1D). These results suggested that ERCE C may consist of multiple cooperative elements, so we deleted COSN2 or both COSN2 and COSN3 in the ΔCOSN1 cell line, generating ΔCOSN12 and ΔCOSN123 (Figs. 1C-vi,vii and EV3xxviii,xxix). ΔCOSN12 resulted in a significant RT delay (Fig. 1D), while deletion of all three OSNs (ΔCOSN123) gave an effect similar to deletion of the entire C element (Fig. 1D,E; Table EV4). We also generated a larger deletion that spans all 3 OSNs (ΔCR13.4k), which also resulted in a delay similar to ΔCOSN123, and ΔC (Figs. 1C-viii,D and EV3xxxi; Table EV4). In summary, all deletions that remove all three OSNs (ΔCR14.6k, ΔCR13.4k, ΔCOSN123) significantly delay RT (Fig. 1D), and these delays are not significantly different from that of ΔC (Table EV4) or from each other (Table EV4). Interestingly, ΔCOSN12 also significantly delayed RT (Fig. 1D) while ΔCR11.8k (which deletes COSN23 and CTSS) did not significantly delay RT (Fig. 1C,D), suggesting that not all combinations of OSN deletions are equivalent and that some OSNs play larger roles than others in different contexts.

Since we were only able to recover one clone of ΔCONS123 with all three deletions in the same allele, we performed four independent Repli-seq experiments starting from separate cultures of this cell line (Fig. EV3xxix). This provided more statistical power, and also allowed us to accumulate sufficient reads to verify that these were the only deletions in the Dppa locus throughout both alleles of this clone (Table EV3), detectable as missing sequences from one allele in the Repli-seq data. To evaluate whether ΔCONS123 resembles ΔC in other contexts, we introduced the full A and B deletions from our previous work into the ΔCOSN123 background, giving rise to cell line ΔA.B.COSN123 (Figs. 1Cix and EV3iv), which fully delayed RT reminiscent of and not significantly different from ΔA.B.C (Fig. 1D–F). We thus conclude that ERCE activity within the "C" element can be accounted for by the three OSN co-bound sites.

## OSN sites contribute to the ERCE activity of the A and B elements

ERCE A, initially discovered with a 3.5 kb deletion, contains one OSN site (AOSN) and one active TSS (ATSS) for the Dppa2 gene that also overlaps an OSN binding site (Fig. 2A,B). Because ΔA had no effect on RT on its own (Fig. EV1Bii,D), and because we had shown that ΔCOSN123 accounts for ΔC, we dissected the "A"

element in the ΔCOSN123 background. Loss of the full activity of AOSN and ATSS is expected to delay RT similar to ΔA.C. First, we introduced either ΔAOSN or ΔA (removing both AOSN and ATSS) into the ΔCOSN123 cell line, creating ΔAOSN.COSN123 and ΔA.COSN123 (Figs. 2Ci–ii and EV3xvi,xi). ΔA.COSN123 significantly delays RT relative to ΔCOSN123 but does not completely recapitulate ΔAC (Fig. 2D; Table EV4), suggesting that the OSN sites in C do not recapitulate the effect of the full C deletion when B is still intact. However, ΔA.COSN123 did cause a significantly greater delay compared to ΔAOSN.COSN123 (Table EV4). In fact, ΔAOSN.COSN123 had no additional contribution to the delay in RT relative to ΔCOSN123 alone (Figs. 2E and EV3; Table EV4). Since ΔA is already relatively small, and since ΔATSS on its own does not delay RT in other contexts (discussed below), it is likely that AOSN and ATSS (which contains an OSN) are cooperative for maintaining RT at ERCE A.

Given our finding of the importance of OSN sites in the C element, we noted that upstream of the A element is another OSN (Z) site that is outside of the original A deletion. Deletion of Z in the context of either ΔAOSN.COSN123 or ΔB.C, producing ΔZ.AOSN.COSN123 and ΔZ.B.C (Figs. 2Ciii,iv and EV3xxxix-xl), respectively, did not delay RT relative to their parental backgrounds (Fig. 2D,F; Table EV4). Consistent with its absence from ERCE A and its presence in ΔA.B.C, which is late-replicating, we conclude that Z is neither necessary nor sufficient for ERCE activity, reminiscent of a strong OSN-containing region we previously designated as "Y" (Fig. EV1 (Sima et al, 2019). Since both Y and Z remain intact in the very late-replicating ΔA.B.C and ΔA.B.COSN123 deletions, the presence of these two OSNs is not sufficient to advance RT. For this reason, we designate the compound elements within ERCEs that contribute to RT as "subERCEs". It will be important to understand what essential component of subERCEs is missing from the Y and Z OSN binding sites (see "Discussion").

ERCE B (~45 kb in size) contains two OSN sites, one ~8 kb upstream (BOSN1) and another ~4 kb downstream (BOSN2) of the TSS for the large (200 kb) Morc1 gene (BTSS) (Fig. 3A,B). BTSS does not contain any detectable OSN binding activity (Fig. 3A,B). Since, like ERCE A, ΔB has a small effect on RT on its own, we dissected the "B" element in the context of other deletions. We first deleted BOSN1 in the context of ΔCOSN123 and ΔA.COSN123 to produce ΔBOSN1.COSN123 and ΔA.BOSN1.COSN123, which produced delays with no statistically significant difference compared to their parental backgrounds (Figs. 3Ci,ii,D and EV3xxii and v; Table EV4). We then deleted BOSN2 in these contexts, producing ΔBOSN12.COSN123 and ΔA.BOSN12.COSN123 (Figs. 3Ciii–iv and EV3xxi,vii). ΔBOSN12.COSN123 was statistically indistinguishable from ΔB.C and

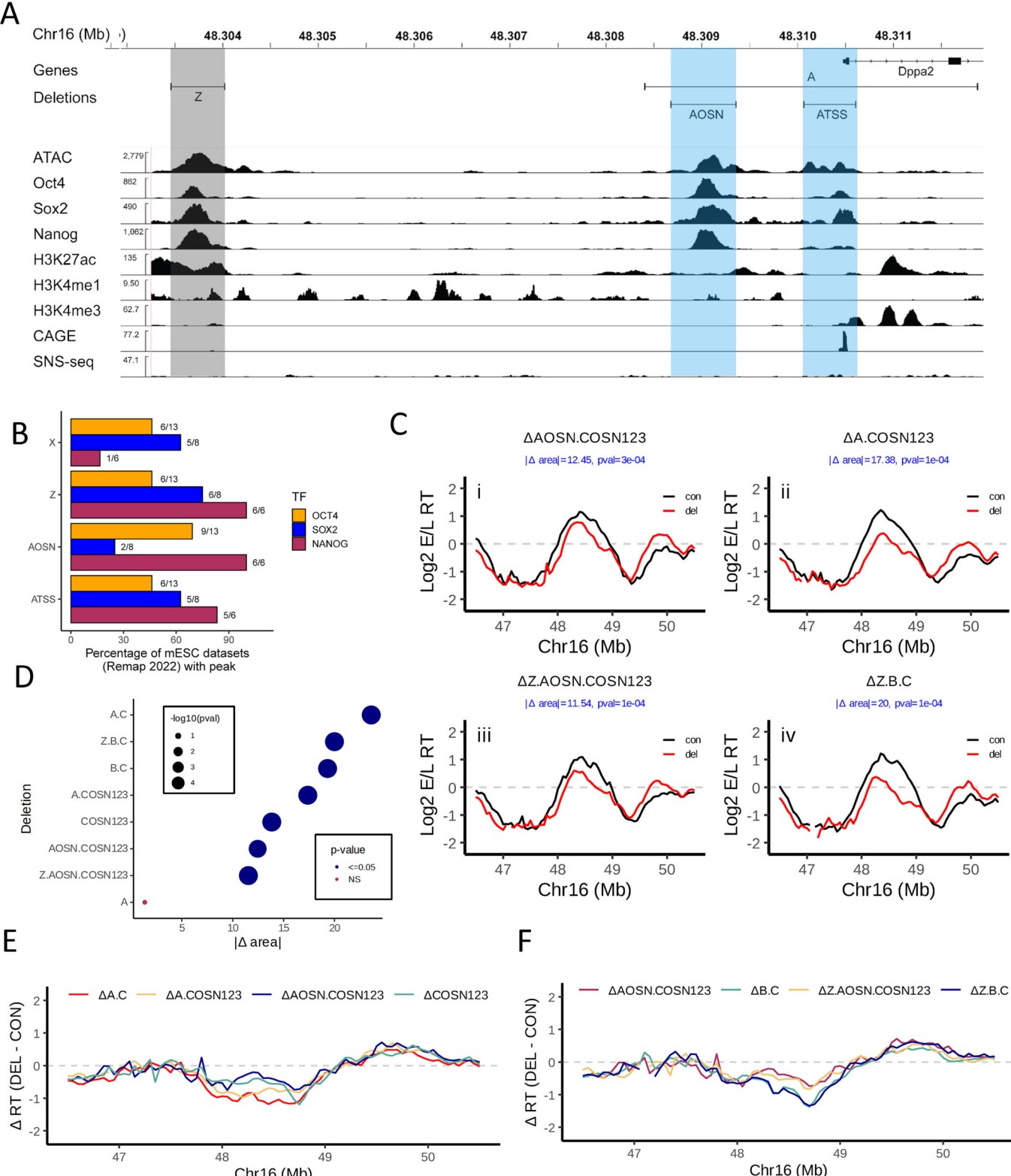

Figure 2. OSN sites contribute to the ERCE activity of the A element.

(A) IGV browser tracks as in Fig. 1A, zooming into the A element. Deletions of the A element are demarcated at the top by horizontal bars above the data tracks. Sites harboring activity are highlighted in blue while sites without activity (Z) are highlighted in gray. (B) Percentage of Remap2022 mESC Oct4, Sox2, and Nanog ChIP-seq datasets with peaks that overlap the deletions generated in the A element as well as an additional upstream site (X, shown in Fig. EV1) that was shown to have no activity in our previous work (Sima et al, 2019). (C) RT profiles of cell lines harboring deletions (red) compared to the homologous unmodified allele (black) in and near ERCE A, presented as in Fig. 1C. (D) |Δ area| between the RT profile curves of the deletions shown in (C) and the significance of their delay in RT compared to the corresponding WT alleles (see "Methods"). Empirical P values are shown. (E, F) Comparison of RT changes [ΔRT (DEL-CON)] in the Dppa domain (E) comparing the deletion of A and AOSN in the ΔCOSN123 and ΔC backgrounds, and (F) comparing the deletion of Z in the ΔAOSN.COSN123 and ΔB.C backgrounds. Source data are available online for this figure.

significantly more delayed than ΔBOSN1.COSN123, demonstrating that, in the absence of COSN123, deletion of BOSN12 can account for the B ERCE RT activity. However, while ΔA.BOSN12.COSN123 was significantly more delayed than ΔA.BOSN1.COSN123 (Fig. 3D; Table EV4), ΔA.BOSN12.COSN123 was still significantly less delayed than either ΔA.B.C or ΔA.B.COSN123 (Fig. 3D,E; Table EV4), indicating that something else is contributing to the promotion of early replication in the absence of all OSN sites with ERCE activity. We also generated the BOSN deletions in the ΔAOSN.COSN123 background, producing ΔAOSN.BOSN1.COSN123 and ΔAOSN.BOSN12.-COSN123 (Figs. 3Cv–vi and EV3xiii–xiv). Consistent with a role for BOSN2 in the activity of ERCE B, ΔAOSN.BOSN12.COSN123 but not ΔAOSN.BOSN1.COSN123, was significantly delayed compared to ΔAOSN.COSN123 (Table EV4). Surprisingly, however, ΔAOSN.-BOSN12.COSN123 was not significantly more delayed than ΔAOSN.-BOSN1.COSN123 (Table EV4) despite the large change in Δarea (12.7 vs. 19), but did show a change in shape of the RT curve that resembles ΔA.BOSN12.COSN123 (Fig. 3Cii–vi). Altogether, our results implicate the OSNs are important for B ERCE activity, but show that their ability to account for B ERCE activity is dependent on the presence of components of the A element via cooperative relationships we do not yet understand.

## Early RT can be maintained in the absence of most transcription

In our previous study, we reported that a triple deletion of A, B and C led to silencing of transcription within the Dppa domain (Sima et al, 2019). However, it remained to be determined whether ERCE RT activity could be uncoupled from their transcription activity. This is particularly important in light of the complex effects of deletions that contained TSSs on early replication and transcription (Sima et al, 2019). To address this question, we generated a series of deletions selectively targeting the three TSSs at A, B, and C followed by E/L Repli-seq to measure RT and BrU-seq to measure nascent transcription (Figs. 4A–F, EV3, and EV4). The CTSS deletion in the WT background, whose nonsignificant effect on RT was described in Fig. 1Ciii,D, had little effect on transcription (Fig. 4A,F). We deleted BTSS in both the WT background and in the ΔCTSS background to generate ΔBTSS and ΔBTSS.CTSS, which both resulted in barely detectable BrU-seq signal from the Morc1 gene (Figs. 4A and EV2) and a reduction in Dppa2 and Dppa4 transcription, but only a modest impact on RT, maintaining early RT (Figs. 4Bi,ii and EV3xxiii,xxiv).

Next, to eliminate transcription from the Dppa2 gene, we deleted ATSS. ΔATSS alone resulted in undetectable BrU-seq signal from the Dppa2 gene and a reduction in Dppa4 with no effect on Morc1 transcription (Figs. 4A and EV4) and a nonsignificant effect on RT (Figs. 4Biii and EV3xvii). Deletion of all three TSSs

(ΔATSS.BTSS.CTSS) resulted in undetectable BrU-seq signal from Dppa2 and large reductions in BrU-seq signal from Morc1 and Dppa4 (Figs. 4A and EV4) with a modest change in RT similar to ΔBTSS.CTSS (Figs. 4Biv,C,F and EV3xviii). Importantly, despite the nearly complete elimination of all transcription in this triple TSS deletion, Dppa domain replication is still occurring in the first half of S phase (Fig. 4Biv). Thus, we conclude that, while transcription plays a role in advancing RT, most transcription is not necessary for early replication at the Dppa domain when subERCEs are intact.

Because ΔBTSS eliminates almost all transcription of the 1 Mb Morc1 gene (Fig. 4A) and ~76% of total BrU-seq signal in the domain, but remains early replicating (Fig. 4Bi,C,F), we generated a BTSS deletion in the ΔAC background, and found that ΔA.BTSS.C resulted in a significantly greater delay than ΔAC (Figs. 4Bv and EV3xviii vs. 3ix and Table EV4), from slightly early-mid S to slightly late-mid S. These data support a role for BTSS in advancing RT when A and C are absent.

Recall from Fig. 3Civ that deletion of all OSNs within all three ERCEs failed to completely delay RT equivalent to ΔA.B.C. Since the BTSS remains in this configuration and is the most consequential TSS deletion for both RT and transcription, we determined whether the BTSS was responsible for both transcription and the mid-late RT in this clone. We found BrU-seq signal was nearly eliminated in ΔABOSN12.COSN123 even though BTSS and CTSS are still present (Figs. 4A,F and EV4). Deletion of BTSS to create ΔA.BOSN12.BTSS.COSN123 did not significantly delay RT further (Fig. 4Bvi,C,D,F and 3EV3vi; Table EV4). These data demonstrate that subERCEs play a role in transcription but when transcription is already crippled by their absence, the BTSS no longer plays a role in RT, suggesting that the role of BTSS is likely through transcription.

Overall, we conclude that subERCEs have roles in both transcription and RT but that when nearly all transcription is eliminated through deletion of the TSSs, they can still maintain early replication, uncoupling these two functions of subERCEs.

## Early RT is independent of where replication initiates

We previously (Sima et al, 2019) aligned the ERCE-containing deletions to available small nascent strand microarray mapping data in mESCs (Cayrou et al, 2015) and reported that one of many origins in the Dppa domain was detected within the left half of the C deletion and none in the A or B deletions (Fig. EV1). Since then, several new mESC replication initiation datasets using various methods have been published, prompting us to re-examine their alignment with the more refined sites of RT activity within ERCEs. The top two data tracks in Fig. 5A align the Dppa locus and its ERCEs with data for High-resolution Repli-seq (Zhao et al, 2020) and Ok-seq (Petryk et al, 2018), which map the first sequences to

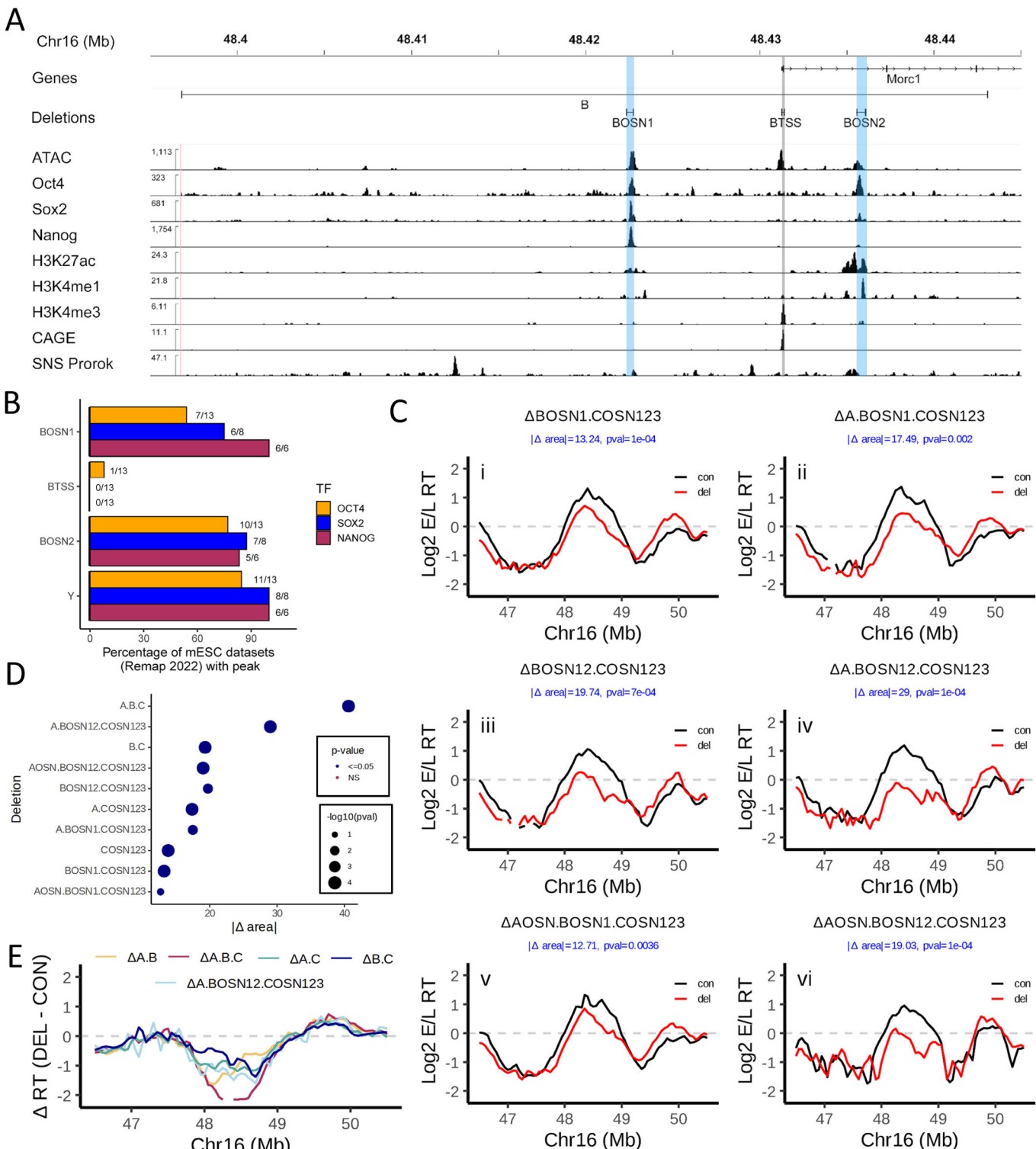

replicate and the population-averaged polarity of replication, respectively, both at ~50 kb resolution. Each detects large regions of dispersed initiation activity termed "initiation zones" (IZs), although Repli-seq IZs score all earliest 50 kb windows above background, while the Ok-seq IZ call is a single site marking the "center of gravity" of surrounding initiation activity (Petryk et al,

2018). In genome-wide data, all mESC Ok-seq IZ calls are early replicating, and align with the early replicating IZs called by High-resolution Repli-seq (Zhao et al, 2020). Indeed, at the Dppa locus, the transition in Okazaki fragment polarity (Ok-seq IZ) is near the center of the earliest DNA synthesis IZ detected by high-resolution Repli-seq. In fact, the IZ called by Repli-seq (Zhao et al, 2020)

**Figure 3.  OSN sites contribute to the ERCE activity of the B element.**

(A) IGV browser tracks as in Figs. 1A and 2A, zooming into the B element. Deletions of the B element are demarcated at the top by horizontal bars above the data tracks. OSNs are highlighted in blue while the TSS is highlighted in gray. (B) Percentage of Remap2022 mESC Oct4, Sox2, and Nanog ChIP-seq datasets with peaks that overlap the deletions generated in the B element as well as an additional site 12 kb downstream of B (Y, not shown in (A)) that was shown to have no activity in our previous work (Sima et al, 2019). (C) RT profiles of cell lines harboring deletions (red) compared to the homologous unmodified WT allele (black) of sites within ERCE B presented as in Fig. 1C. (D) |Δ area| between the RT profile curves of the deletions shown in (C) and the significance of their delay in RT compared to the corresponding WT alleles (see "Methods"). Empirical *P* values are shown. (E) Comparison of RT changes [ΔRT (DEL-CON)] in the Dppa domain comparing ΔA.BOSN12.COSN123 to ΔA.B.C and pairwise ERCE deletions. Source data are available online for this figure.

extends from one border of strong Ok-seq polarity to the other border of opposite polarity (Fig. 5A), coinciding with a broad region of reduced Okazaki fragment polarity characteristic of dispersed initiation activity (red dashed box in Fig. 5A), and encompassing all three ERCEs A, B and C. Together, these two very different methods provide high confidence that initiation at the Dppa locus is distributed throughout the entire region encompassed by the ERCEs.

Also shown in Fig. 5A are three recently published mESC small nascent strand sequencing (SNS-seq) mapping datasets (Jodkowska et al, 2022; Pratto et al, 2021; Prorok et al, 2019), which map initiation sites to kilobase resolution. One of these datasets (SNS-seq 1) detects peaks throughout the IZ, including two peak calls within the left half of C and one in the left side of B. The other two datasets (OriSDS and SNS-seq 2) detect 0 and 2 peaks in the entire Dppa domain, none of which reside within the original large ERCE-containing deletions (Fig. 5A). Importantly, none of the SNS-seq datasets detected peaks within 10 kb of any of the subERCEs. Together with the Repli-seq and Ok-seq data, the most parsimonious conclusion is that origin activity in the Dppa locus is distributed throughout the domain, with some sites firing at efficiencies above the threshold of SNS-seq peak detection. To evaluate the importance of efficient sites of initiation for Dppa domain RT, we aligned the position of a large (~245 kb) deletion made in our previous work (Sima et al, 2019), that had no detectable effect on RT (Fig. 5B). This deletion removes all DNA between the A and C elements, encompassing the majority of detectable SNS activity throughout the Dppa domain. Thus, initiation frequencies must necessarily be redistributed in this deletion to the remaining flanking regions; this suggests RT is regulated independently of the choice of sites used to initiate replication.

## Discussion

*Cis*-acting elements controlling large-scale (>hundreds of kb) chromosome structure and function have proven elusive. We recently identified ERCEs as *cis*-elements necessary for early RT, and found that the same chromosome segments harboring ERCEs were also necessary for domain-confined transcription, 3D architecture and A/B compartmentalization. Most of our deletions were over 30 kb, making it difficult to determine whether early replication activity could be uncoupled from these other activities. Here, we have performed a series of fine deletions within the three ERCES residing in the ~400 kb Dppa2/4 gene-containing domain. We demonstrate that a set of six relatively small (140–3483 bp; 7384 kb total) deletions can mostly recapitulate the very early to very late replication timing switch observed with the larger deletion series (which deleted 88.7 kb total).

These six deletions contain seven sites that are bound by diverse master transcription factors, which we term subERCEs. Although the three subERCEs in ERCE C account for its RT activity in most deletion series, deletion of all seven subERCEs does not render the entire domain homogenously very late-replicating, such as what we see with ΔA.B.C. At present, we do not know what alternative activity is contributing to the remaining late-middle RT in these clones. Deletion of all subERCEs eliminates almost all transcription in the locus, despite retaining all domain TSSs. On the other hand, deleting three TSSs (ΔATSS.BTSS.CTSS), leaving 6/7 subERCEs intact (the seventh is ATSS, which may also be a subERCE), also eliminates almost all transcription, but the Dppa locus replicates early in this clone. Thus, subERCEs have roles in both transcription and RT but, when nearly all transcription is eliminated through deletion of the TSSs, they can still maintain early replication, uncoupling these two functions of subERCEs. Note that a 140 bp deletion of only the BTSS substantially reduces transcription of the long Morc1 gene, with a significant effect on RT. However, when transcription was already reduced by deleting all 7 subERCEs, further deletion of the BTSS had no additional effect on RT, suggesting that the role of BTSS in RT is strictly through transcription, distinguishing it from subERCEs. Together, we conclude that subERCEs are compound elements consisting of multiple independent sites that maintain early RT through both transcription-dependent and independent roles. Thus, changes in transcription factors during cell fate changes can alter RT without affecting transcription and may cooperate with transcription to elicit robust changes in RT.

## ERCEs consist of multiple smaller "subERCEs"

Our original deletion study (Sima et al, 2019) found that very early replication requires at least two ERCEs. Here, we show that each of the ERCEs consists of multiple smaller elements, identified by OSN co-binding. However, it is clear that the OSN TFs themselves are not sufficient for ERCE RT activity, since two additional strong OSN sites (Y and Z) remain intact in all of our late S-phase replicating deletion clones. Since the term OSN is thus misleading with respect to RT activity, we have introduced the term "subERCEs" for elements with non-transcriptional roles in RT. Importantly, both motif analysis and ChIP data identify a large number of TFs that bind to subERCEs (Fig. EV5A–C). In fact, it has been shown that the diversity of TF-binding sites is more important for predicting enhancer activity than the specific identity of TFs (Singh et al, 2021). However, X, Y, and Z and even CTSS also contain many TF-binding sites (Fig. EV5B), so there is no obvious combination of TFs that stands out as distinguishing subERCEs, and the number of elements in our study is too small to find a TF signature unique to subERCEs. Figure EV5D shows recently published high-resolution H3K27ac HiChIP and Micro-C data for the Dppa locus in mESCs (preprint: Jusuf et al,

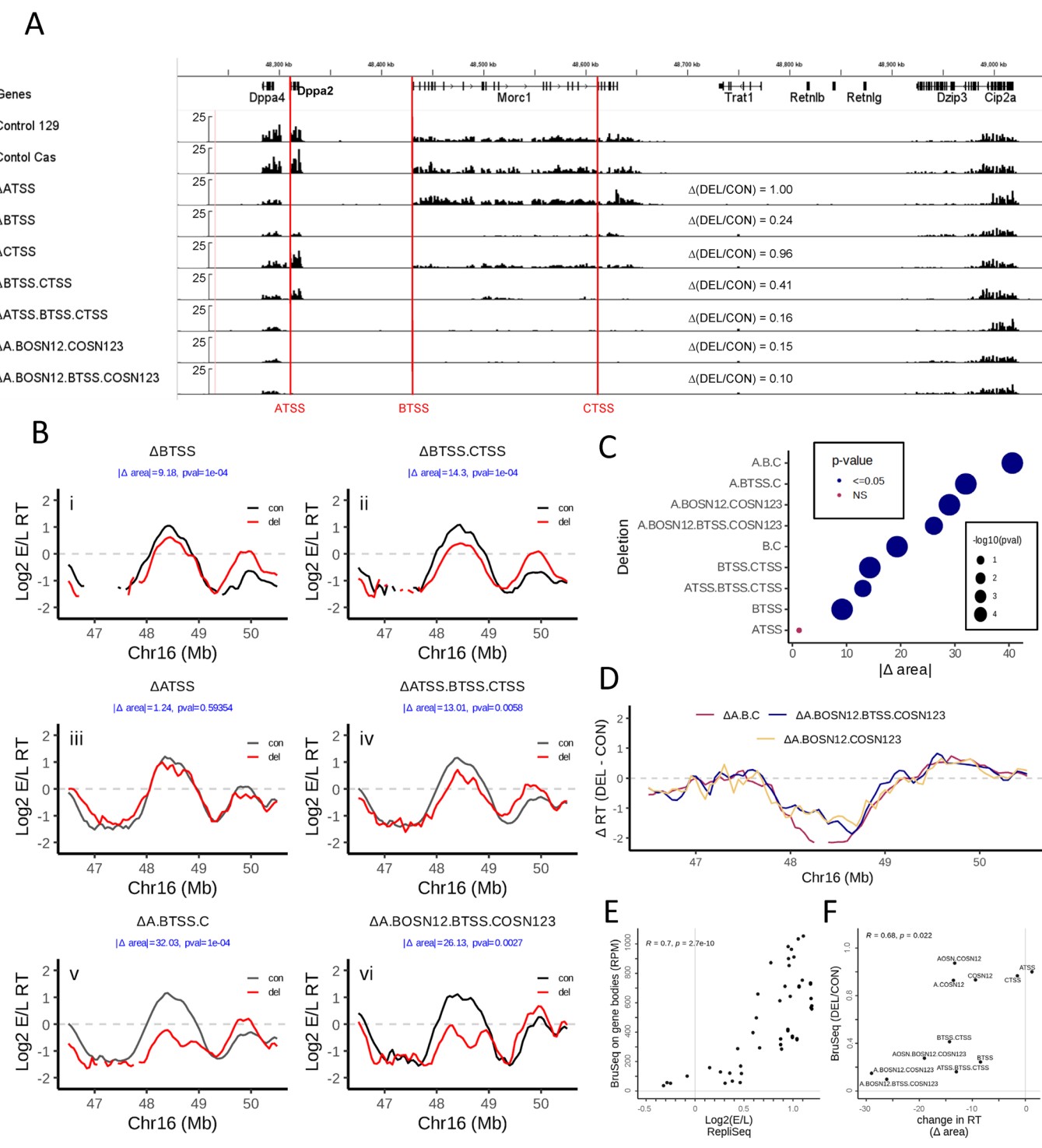

**Figure 4. Most transcription is not required for early RT of the *Dppa2/4* domain.**

(**A**) Representative IGV Bru-seq tracks with TSS deletions indicated at red vertical lines. Tracks of all individual replicate BrU-seq datasets generated in this study are shown in Fig. EV4. (**B**) RT profiles of cell lines harboring deletions (red) compared to the homologous unmodified WT allele (black) of sites within ERCE B presented as in Fig. 1C. For plots in which the control (con) is gray, at least one of the clones has a deletions on both alleles and the control is an average of all control alleles in all clones in this study. (**C**) |Δ area| between the RT profile curves for the deletions shown in (**B**) and the significance of their delay in RT compared to their corresponding control (see "Methods"). (**D**) Comparison of RT changes [ΔRT (DEL-CON)] in the Dppa domain comparing ΔA.BOSN12.BTSS.COSN123 to ΔA.BOSN12.COSN123 and ΔA.B.C. Empirical p values are shown. (**E**, **F**) Scatter plots of all clones for which both RepliSeq and BruSeq were performed, showing (**E**) the correlation between the average RT of Dppa domain map coordinates 48.2–48.6 Mb for each clone and the corresponding BruSeq signal over all three gene bodies (Dppa2/4 and Morc1), and (**F**) the correlation between the relative change in BruSeq (DEL/CON) and the change in RT (Δ area).

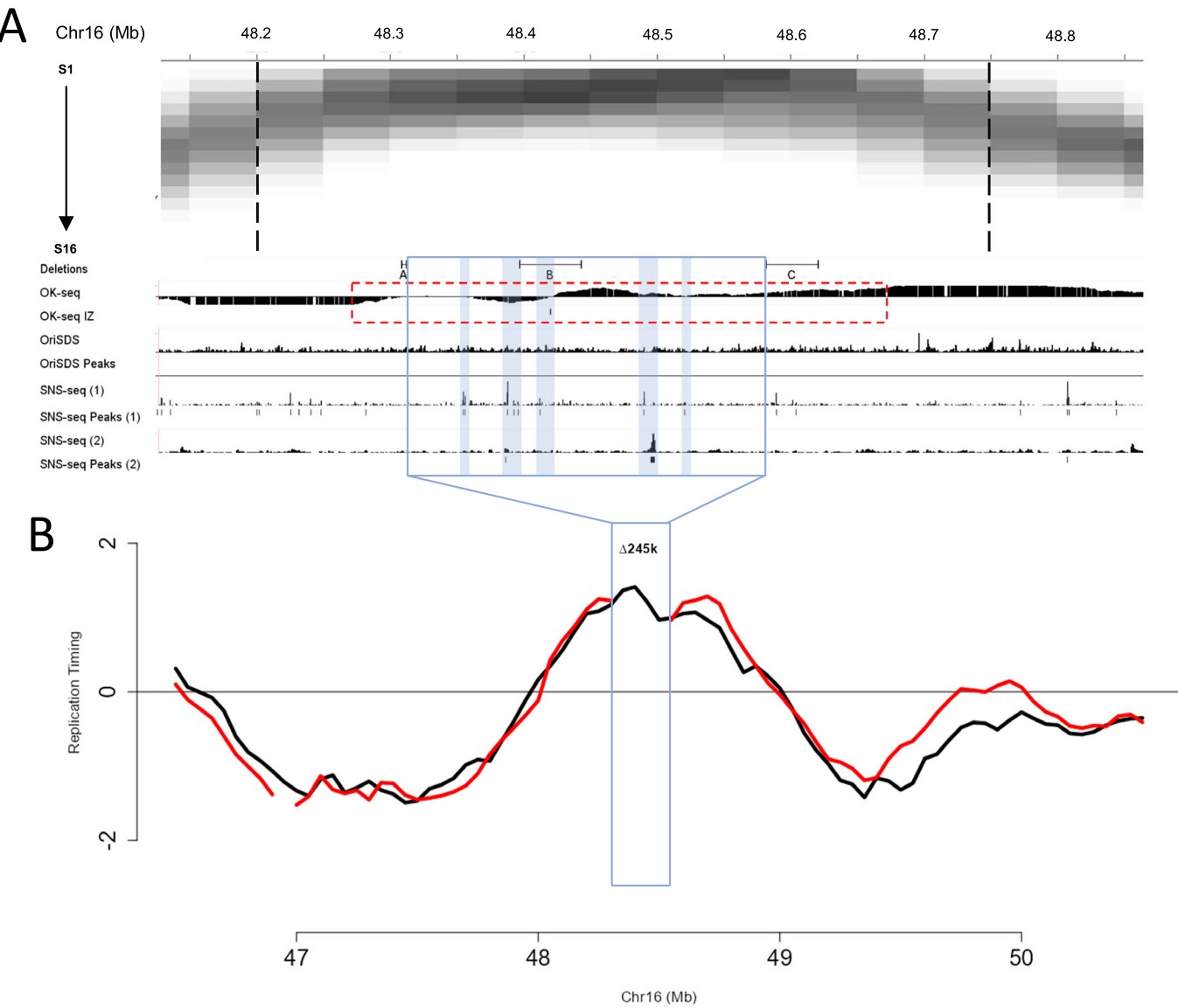

**Figure 5. ERCEs and replication origins.**

(A) IGV tracks showing: (Top) a heatmap of read coverage in 50 kb genomic windows from high-resolution Repli-seq of the *musculus* allele in F121-9 mESCs, with the IZ call indicated by the vertical black dashed lines, which identifies the locally earliest 50 kb bins with read coverage significantly above background, (Zhao et al, 2020). (Middle) Deletions and signal tracks followed by peak calls for Okazaki fragment sequencing (Ok-seq) data (Petryk et al, 2018). The "IZ Peak" calls in Ok-seq data represent the point where the population average of fork directions switches polarity. The red dashed box represents the region of reduced unidirectional replication (increased bidirectional replication), indicative of origin firing. (Bottom) Signal tracks followed by peak calls for three small nascent strand sequencing (SNS-seq) datasets. OriSDS is a revised SNS method in which only bidirectional small nascent strands are scored as peaks (Pratto et al, 2021). SNS datasets (1) and (2) are from (Prorok et al, 2019) and (Jodkowska et al, 2022), respectively. (B) Log2(E/L) Repli-seq of a clone with a 245 kb deletion encompassing most of the efficient SNS origin peak calls (Sima et al, 2019). Blue box extending up into (A) expands the deleted region to show positions of SNS and Ok-seq IZ peak. Source data are available online for this figure.

2025; Kraft et al, 2022). Consistent with our previous capture Hi-C data (Sima et al, 2019), interactions are seen between ERCEs A and C and between B and C, but not between A and B. Consistent with the activity of subERCEs being linked to their interactions, interactions are enriched at the subERCE sites within the ERCEs, albeit in all cases, TSSs are in close proximity to the subERCEs. Interestingly, Y and Z show no significant interactions with any subERCE in either the HiChIP or micro-C data, and X only has one contact with ERCE A. Thus, it is possible that inter-ERCE interaction distinguishes subERCEs. A recent study demonstrated that superenhancers contain bioinformatically equivalent but functionally distinct "facilitators" that potentiate enhancer activity dependent upon their positions within the superenhancer (Blayney et al, 2023), raising intriguing parallels to subERCEs within ERCEs. Facilitators were identified using a whole-locus synthetic biology-enabled engineering strategy designed to overcome the difficulties of introducing multiple independent mutations in a single allele. Novel approaches such as these will be needed to elucidate mechanisms by which subERCEs maintain early replication, potentially coordinating changes in RT with transcription during differentiation.

## ERCEs and transcription

We demonstrate that subERCE RT activity can maintain early replication independent of its roles in transcription, uncoupling these two identified activities of ERCEs. Even after all ERCE-linked transcriptional activity (all the transcription eliminated in ΔA.B.C) and 90% of total transcriptional activity throughout the Dppa domain, is eliminated by deleting ERCE-resident TSSs, the domain remains early replicating. However, our results also suggest a role for transcription in the robust maintenance of Dppa domain RT. Deletion of subERCEs severely diminishes transcription, suggesting they do play roles as transcriptional enhancers, and we have no mutants that are late-replicating while retaining substantial levels of transcription. It has been shown that high levels of transcription, or transcription of long genes can be sufficient but not necessary to drive early RT during differentiation (preprint: Vouzas et al, 2025), suggesting something other than transcription, such as ERCEs, function when transcription is not activated. Moreover, transcription of small, lowly transcribed genes such as Dppa2/4 may not influence RT at all (Blin et al, 2019; preprint: Vouzas et al, 2025). Our work shows that, at the Dppa locus, the BTSS deletion, which eliminates most domain transcription, including the long Morc1 gene, causes a significant delay in RT, but is not enough to erase early replication, even after additional deletion of ATSS and CTSS, so transcription contributes to how early the Dppa locus is, but is not necessary for early replication. Thus transcription is one of at least two independent mechanisms maintaining Dppa RT. The effect of transcription on RT may be direct (preprint: Vouzas et al, 2025). Alternatively, transcription could result in alterations in the positions of the replicative MCM helicase complexes that are known to be cleared from the bodies of active genes (Chen et al, 2019; Hu and Stillman, 2023; Lichauco et al, 2025; Liu et al, 2021; Sasaki et al, 2006; Scherr et al, 2022). Thus, silencing transcription could result in a broad distribution of origin firing across the large Morc1 gene body, each of which is equally early firing but used infrequently, so that any given genomic bin appears to replicate later in ensemble data. Indeed, this model could explain why the 245kb deletion that removes most of the Morc1 gene has no effect on RT (Fig. 5), while the 140-bp BTSS deletion significantly delays RT (Fig. 3); the 245kb deletion removes most of the Morc1 gene body, which may re-focus MCM and initiation to a smaller genomic region. Regardless of the mechanism, the most parsimonious interpretation of our results is that promoters, such as BTSS, advance RT strictly by driving transcription, while subERCEs advance RT both through their role in transcription, likely functioning as enhancers or facilitators, and through a transcription-independent mechanism. Until we understand mechanism, it will remain difficult to bioinformatically identify subERCEs.

## ERCEs and replication origins

Although replication initiates at sites dispersed throughout the Dppa locus, RT is not linked to any specific sites of initiation. This is consistent with a body of literature uncoupling RT from origin specification, discussed in depth elsewhere (Rivera-Mulia et al, 2016). Indeed, the role of any site-specific origin of replication in chromosome biology has been difficult to substantiate. There is evidence that altered origin usage can influence the frequency of trinucleotide repeat expansion by altering the polarity of replication forks passing through the repeats (Gerhardt et al, 2014). There is also evidence of an early-to-late RT change upon deletion of a

strong SNS-seq peak in mouse B cells (Malzl et al, 2023), although ReMap2022 ChIP data reveals that the deleted site binds diverse TFs, including B cell master regulators such as Pax5, and so could contain one or more subERCEs. Here, we demonstrate that we can delete the most efficient origins within an initiation zone, leaving two ERCEs behind, with no detectable effect on RT.

To the non-expert, the discordance between origin peak calls in Fig. 5 may seem surprising. In fact, it is typical of most genomic regions in mammalian cells, particularly developmentally regulated domains (Besnard et al, 2012; Dileep et al, 2015; Takebayashi et al, 2012b), likely due to a combination of technical noise and the biological background of dispersed initiation. The usage of sites of replication initiation in mammalian cells is highly stochastic (Bechhoefer and Rhind, 2012; Carrington et al, 2025; Hyrien, 2015; Wang et al, 2021). Single-molecule methods show that 20% of origins are organized in clusters averaging 45 kb termed Initiation Zones (IZs), within which replication initiates at one or a few of many more potential sites. The remaining 80% of initiations are scattered throughout the genome outside of the IZs and fire at an undetectably low frequency (Wang et al, 2021; Carrington et al, 2025). This has made origin mapping very difficult, as single-molecule methods are challenging and population-based methods that achieve kilobase resolution, such as SNS-seq (Jodkowska et al, 2022; Pratto et al, 2021; Prorok et al, 2019) and initiation site sequencing (Ini-seq; (Guilbaud et al, 2022) detect only those initiation sites with sufficient efficiency to give a signal above noise. There are, however, some exceptionally efficient and reproducibly detectable "core origin" sites (Akerman et al, 2020), which mainly reside within gene-rich domains replicated very early across cell types. By contrast, lower resolution methods Ok-seq (Petryk et al, 2016) and High-resolution Repli-seq (Zhao et al, 2020), which detect sites where many origins are clustered into IZs, give high concordance between replicates and even between these two very different data types (Fig. 5 and Zhao et al, 2020). In summary, IZs are broad regions of chromatin where replication initiation is enriched, and the entire Dppa replication domain can be considered one (or possibly two) large IZ. ERCEs, on the other hand, are discrete elements required for large-scale domain structure and function. Taken together, we favor a model in which ERCEs establish a subnuclear environment that promotes initiation stochastically at available sites within their realm of influence, the replication domain (Fig. 6; Dimitrova and Gilbert, 1999; Gilbert, 2001; Sima et al, 2019).

## Co-opting TFs as a means to regulate RT during development

Our results strongly implicate master transcription factor binding sites in the developmental control of RT. Coupled transcription and RT activities of a subset of enhancers suggest one possible reason why RT and transcriptional regulation have been so hard to untangle, and a partial explanation for why there are so many more cell-type specific TF-binding sites than there are essential transcriptional enhancers (Lo et al, 2022; Rothenberg, 2022). Our results also suggest that cell-type-specific ERCEs could respond to changes in transcriptional regulatory networks during differentiation to trigger replication timing changes, thus remodeling the epigenomic composition of their resident domain to create an environment more conducive for transcription (Klein et al, 2021; Rivera-Mulia et al, 2019a). Our results are also consistent with a role for transcription in advancing RT, as reported by others (Blin et al, 2019; Therizols et al, 2014; preprint:

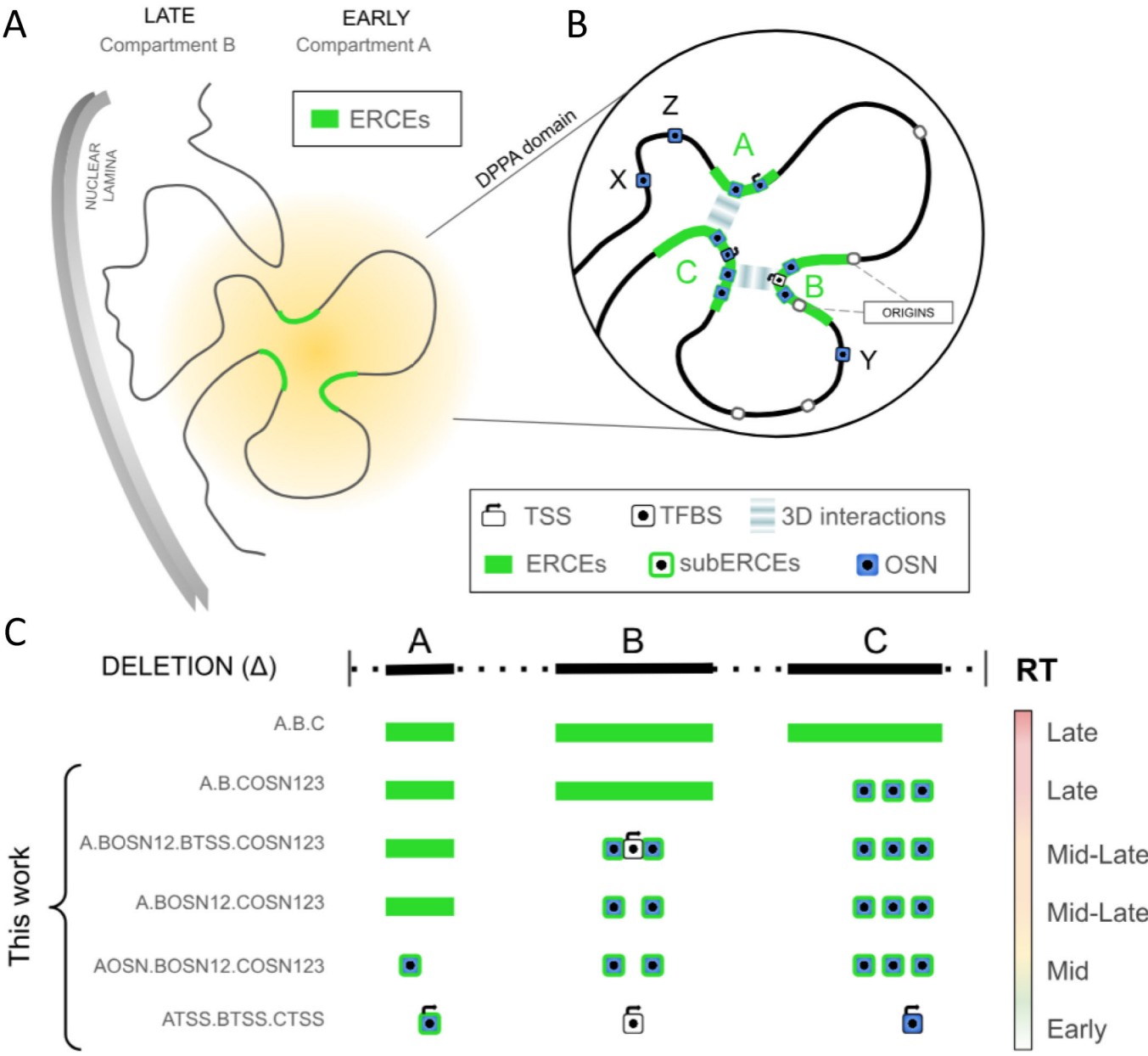

**Figure 6. Uncoupling activities within ERCEs.**

Working model for ERCE function, illustrating the various activities of candidate regulatory elements subject to deletion analysis in this work, while retaining the basic tenets of our original proposed model (Sima et al, 2019). (A) Original version of the model adapted from Sima et al, 2019, showing a hypothetical structure of the Dppa replication domain, in which interactions between ERCEs seed the assembly of a microenvironment (yellow halo) that results in interactions with the A chromatin compartment and increases the local probability of origin firing. (B) Revised model showing the positions and interaction of all dissected elements in this study (marked by boxes) and classified according to function. Bubbles represent stochastically used replication origins within an initiation zone that aligns with ERCE B. Interactions occur between subERCEs A and C and B and C, but are rarely detected between A and B. (C) Schematic representation of relevant compound deletions and their resulting effects on RT, highlighting the contribution of this work towards a detailed understanding of ERCE function.

Vouzas et al, 2025), and suggest that these mechanisms can act independently. Potentially then, either transcription or ERCEs could initiate RT advances during cell fate specification, while the alternative mechanism could serve as a positive feedback to drive stable transitions in epigenetic state in either temporal order. Testing this model in mammalian cells will require further understanding of the mechanisms by which ERCEs function.

Figure 6 illustrates a speculative model for ERCE function, modified from its original version (Fig. 6A; Sima et al, 2019) to include the discovery of TF-rich subERCEs, as well as recently published origin mapping and Micro-C/HiChIP data (Fig. 6B and EV5D). Figure 6C summarizes key observations in this report. Here we demonstrate that ERCEs consist of multiple subERCEs, which are sites of diverse cell-type-specific TF binding. The subERCEs

account for most of the RT advancing activity of ERCEs. ERCEs were originally discovered by virtue of their CTCF-independent interactions (Sima et al, 2019). We show here that ERCE interactions are focused on the more localized subERCEs (Fig. EV5D), while Y and Z do not interact with subERCEs. Interestingly, synthetic sequences that can advance RT in chicken DT40 cells, when inserted at distant sites, have been shown to interact in 3D space and when doing so they synergize in promoting early replication (Brossas et al, 2020). We propose that subERCEs create a subnuclear microenvironment, potentially through (micro)phase separation (preprint: Goel et al, 2024), that promotes initiation of replication within the structurally organized replication domain. Formation of a microenvironment can also explain why ERCE deletions tend to result in shifts of peak early replication to the general region of remaining ERCEs that may promote more localized microenvironments. In this model, the switch to late replication of the Dppa domain upon differentiation would accompany the downregulation of the pluripotency-specific TFs that bind to subERCEs. Given that we have validated ERCEs in two other domains (Sima et al, 2019), it is likely that the mechanisms functioning at the Dppa domain also function in other developmentally regulated replication domains.

This model for subERCE activity is purposely reminiscent of binding sites for the Fkh1/2 TFs in budding yeast, which reside near a set of early firing origins and promote early RT by dimerization-mediated interactions, independently of their role in transcription (Ostrow et al, 2017). In fact, Oct4 and Nanog, as well as Yamanaka factor Klf4, have been shown to mediate chromatin looping (Choi et al, 2022; de Wit et al, 2013; Di Giammartino et al, 2019), and several studies have shown that master TFs form 3D hubs to regulate cell identity (Hu et al, 2023; Liu et al, 2023; Madsen et al, 2020; Wang et al, 2022a; Wang et al, 2022b; Winick-Ng et al, 2021). FoxP1, a mammalian TF that possesses the same dimerization motif as Fkh1/2 (Ostrow et al, 2017), is highly expressed in ESCs and required for pluripotency (Gabut et al, 2011). Moreover, both Sox2 and Klf5 interact with the pluripotency-specific isoform of FoxP1 (Göös et al, 2022), which contains a self-dimerization motif homologous to that of budding yeast Fkh1/2. Thus, mechanisms regulating constitutive replication timing in budding yeast may inform us as to how RT changes occur during mammalian cell fate transitions. It is now of high interest to determine whether one or more of the TFs bound to subERCEs mediates interactions between ERCEs, and if so, whether these interactions regulate RT independently of their role in transcription.

# Methods

### Reagents and tools table

| Reagent/resource | Reference or source | Identifier or catalog number |
| --- | --- | --- |
| **Experimental models** | | |
| F121-9 mESC | Rivera-Mulia et al, 2018 | |
| **Recombinant DNA** | | |
| pX458 | Addgene | Plasmid #48138 |
| **Antibodies** | | |
| Anti-BrdU | BD | 555627 |

| Reagent/resource | Reference or source | Identifier or catalog number |
| --- | --- | --- |
| **Oligonucleotides and other sequence-based reagents** | | |
| gRNA cloning oligos | This study | Table EV1 |
| PCR primers | This study | Table EV1 |
| **Chemicals, enzymes, and other reagents** | | |
| P3 Primary Cell 4D Nucleofector X Kit | Lonza | V4XP3024 |
| XFect Transfection Reagent | Takara | 631318 |
| BrdU | Sigma-Aldrich | B5002 |
| NEBNext Ultra Library DNA Prep Kit for Illumina | New England Biolabs | E7370 |
| BrU | Sigma-Aldrich | 850187 |
| Direct-Zol RNA Miniprep Plus Kit | Zymo | R2070 |
| NEBNext Ultra II RNA First Strand Synthesis Module | New England Biolabs | E7771 |
| NEBNext Ultra II Directional Second Strand Synthesis Kit (NEB) | New England Biolabs | E7550 |
| NEBNext Ultra II DNA Library Prep Kit for Illumina | New England Biolabs | E7645 |
| **Software** | | |
| Cutadapt | Martin, 2011 | |
| Bowtie2 | Langmead and Salzberg, 2012 | |
| SNP-split | Krueger and Andrews, 2016 | |
| Samtools | Li et al, 2009 | |
| Bedtools | Quinlan and Hall, 2010 | |
| R | https://www.r-project.org/ | |
| preprocessCore | Bolstad, 2023 | |
| Custom replication timing sequencing data processing scripts | This study | https://github.com/jlt3/ERCEs_2023 |
| Custom replication timing analysis scripts | This study | https://github.com/ay-lab/SignifRT |
| FIMO | Grant et al, 2011 | |
| **Other** | | |
| ATAC-seq | King and Klose, 2017 | GSE87819 |
| Oct4 ChIP-seq | King and Klose, 2017 | GSE87820 |
| Sox2 ChIP-seq | King and Klose, 2017 | GSE87820 |
| Nanog ChIP-seq | King and Klose, 2017 | GSE87820 |
| H3K27ac ChIP-seq | He et al, 2020 | ENCFF646XOT |
| H3K4me1 ChIP-seq | He et al, 2020 | ENCFF016YZA |
| H3K4me3 ChIP-seq | He et al, 2020 | ENCFF6116SQ |
| CAGE | Noguchi et al, 2017 | |
| SNS-seq (1) | Prorok et al, 2019 | GSE126477 |
| SNS-seq (2) | Jodkowska et al, 2022 | GSE131699 |

| Reagent/resource | Reference or source | Identifier or catalog number |
|---|---|---|
| OriSDS | Pratto et al, 2021 | GES148327 |
| OK-seq | Petryk et al, 2018 | GSE117274 |
| High-Res. Repli-seq | Zhao et al, 2020 | GSE137764 |
| E/L Repli-seq | Sima et al, 2019 | GSE114139 |
| ReMap2022 (ChiPseq) | Hammal et al, 2022 | |
| LoopCatalog H3K27ac HiChIP | Kraft et al, 2022; Reyna et al, 2024 | GSE150906 |
| micro-C | Jusuf et al, 2025 | GSE286495 |
| Rad21 ChIP-seq | Cattoglio et al, 2019; Hansen et al, 2017 | GSE90994 |
| HOCOMOCOv11 Mouse Motif database | Kulakovskiy et al, 2018 | |

## Methods and protocols

### Cell culture

mESCs were cultured, maintained, and passaged as previously described (Sima et al, 2019). Briefly, cells were cultured on 0.1% gelatin-coated dishes in serum-free 2i/LIF media.

## Cell line engineering and genotyping

Cells were transfected using P3 Primary Cell 4D Nucleofector X Kit (Lonza, V4XP3024) or XFect Transfection Reagent (Takara, 631318) with two px458 (Addgene, 48138) plasmids containing gRNA sequences targeting the deletion breakpoints. Three days after transfection, GFP-positive cells were single-cell sorted by FACS into 96-well plates. Ten to fourteen days after sorting, colonies were picked and expanded into duplicate 96-well plates and PCR screened. Primers upstream and downstream of the breakpoints are used to identify deletion mutants, and for large (>2 kb) deletions, primers within the deleted sequences are used to identify heterozygous mutants. Deletion PCR fragments were then Sanger-sequenced to determine the haplotype of the deletion allele, which was then verified by NGS. The sgRNA and PCR primer information can be found in Table EV1. Deletion coordinates and sgRNAs used for each deletion can be found in Table EV2. Individual cell line genotypes as determined by Sanger sequencing and NGS analysis can be found in Table EV3. Cell line authentication was done by analyzing the whole genome repli-seq data, and all lines are free of mycoplasma.

## Repli-seq library preparation and data analysis

Repli-seq was performed as previously described (Marchal et al, 2018). Briefly, cells were labeled with 100uM BrdU (Sigma-Aldrich, B5002) for 2 h and then fixed in 75% ethanol. Fixed cells were then sorted by FACS into early-S and late-S fractions based on propidium iodide staining of DNA. DNA was then purified from each fraction, sheared using Covaris ME220, and used to construct Illumina sequencing libraries with NEBNext Ultra Library DNA Prep Kit for Illumina (NEB, E7370). BrdU-labeled nascent DNA library fragments were then enriched by immunoprecipitation with anti-BrdU antibody (BD, 555627). IP products are then PCR amplified and indexed. A minimum of 30 million reads/read pairs per library was requested. Raw sequencing data were first quality and adapter trimmed using TrimGalore (https://github.com/FelixKrueger/TrimGalore) (Martin, 2011) and then aligned to the reference genome with bowtie2 (Langmead and Salzberg, 2012). Reads were aligned to a custom N-masked mm10 genome for subsequent haplotype phasing with SNP-split (Krueger and Andrews, 2016). Aligned reads were then quality filtered and duplicate reads removed using Samtools (Li et al, 2009). Processed aligned reads were then counted in 50 kb windows across the entire genome for both early- and late-fraction using bedtools (Quinlan and Hall, 2010) and then a log2 ratio of early-to-late read counts for each window was calculated to generate raw timing files. Raw E/L data were then scaled with R, quantile normalized with R package preprocessCore (Bolstad, 2023) and loess-smoothed in 300 kb windows to generate the final replication timing profiles used for plotting and analysis. Scripts for processing the data can be found at Github (https://github.com/jlt3/ERCEs_2023). All individual datasets are plotted in their respective groups in Fig. EV3.

## Statistical analysis of RT changes

To quantitatively measure the effect of a deletion in delaying the replication timing of the DPPA domain, we implemented a randomization-based statistical approach. We simply ask the question of whether the delay in RT (computed as the area between curves representing control and deletion alleles) observed from a given real deletion is significantly larger than expected by chance. First, we calculated the area under the averaged $\Delta$ RT curve (deletion—control) for each target deletion in the DPPA domain (chr16 47.5 Mb → 49.5 Mb) using the AUC function in R. For the clones without a WT internal control (homozygous deletions), we averaged the RT profiles from all available WT alleles from all the clones in this work to generate the aggregate WT curve. To estimate random fluctuations in the RT data between the two alleles, we built a background model by randomly sampling the genome to obtain ten thousand regions size-matched to the query (2 Mb DPPA domain in this case), excluding previously characterized genomic regions with allele-specific RT in the 129/cas hybrid line (Rivera-Mulia et al, 2018). We utilized a $P$ value cutoff of 0.05 to determine which deletions significantly delay the RT of the DPPA domain. Similarly, to compare between two deletions, we employed this approach to test whether the change in RT of two distinct deletions is statistically equivalent, by comparing the $\Delta\Delta RT = \Delta RT1 - \Delta RT2$ with the same randomization-based framework as described above. We developed a generic code that can assess the significance of RT change for any query region with options to allow choosing sampling size, statistical test, and consider blacklisted regions. We made these scripts available on Github: https://github.com/ay-lab/SignifRT. The statistics for all datasets and a graphical explanation of the method used can be found in Fig. EV2. All pairwise statistical comparisons can be found in Table EV4.

## Bru-seq library preparation and data analysis

Bru-seq was performed as previously described (Bedi et al, 2020; Paulsen et al, 2014). Briefly, cells were labeled with 2 mM Bru (Sigma-Aldrich, 850187) for 30 min and then total RNA collected

using Direct-Zol RNA Miniprep Plus Kit (Zymo, R2070). Bru-labeled nascent RNA was then enriched 47 using anti-BrdU antibodies conjugated to magnetic beads. Nascent RNA was then fragmented at 85 °C for 10 min and converted to cDNA using NEBNext Ultra II RNA First Strand Synthesis Module (NEB, E7771) and NEBNext Ultra II Directional Second Strand Synthesis Kit (NEB, E7550) according to the manufacturer's instructions. cDNA fragments were then constructed into libraries using NEBNext Ultra II DNA Library Prep Kit for Illumina (NEB, E7645). In all, 100 M reads/read pairs per library were requested. Raw sequencing data were processed and aligned like in E/L repliseq, except duplicate reads are retained (Paulsen et al, 2014). RPM-normalized basepair coverage tracks were generated from the processed alignment files with bedtools and visualized in bigwig file format using IGV. All Bru-seq performed and analyzed in this study can be found in Fig. EV4.

## Bru-seq quantification and Bru-seq vs RT analysis

RPM-normalized BruSeq signal at 1 kb resolution was averaged across gene bodies of Morc1, Dppa2, and Dppa4 for each available clone. Data available in Table EV5. To compare against RT, the log2(E/L) of the DPPA replication domain was averaged across replicates of each genetically distinct clone for which both RepliSeq and BruSeq were performed. The change in BruSeq was computed as the fraction of signal observed in all deletion alleles over the average signal of all available control alleles for the corresponding genome. ΔRT was calculated as described above. Pearson correlations were computed using the stat_cor() function in R.

## TFBS analyses

A bedfile containing all of the peak calls from the ReMap2022 database (https://remap2022.univ-amu.fr/) for mouse samples was downloaded and filtered for the "mESC" biotype, and all other peak calls were removed. We then aligned the peak calls for individual transcription factors (Oct4, Sox2, Nanog in Figs. 1–3; all factors for Fig. EV5) to the genomic coordinates of our deletion sites and other sites of activity within the Dppa domain. We only counted one peak call from one dataset per site, and as such, we only report the number of datasets that have at least one peak call in a given site. Motif search analysis on the deleted sequences was performed with FIMO (Grant et al, 2011) against the HOCOMOCOv11 Mouse Motif database (Kulakovskiy et al, 2018) with 1% FDR threshold.

## Data availability

The sequencing data from this publication have been deposited in the Gene Expression Omnibus (GEO) (https://www.ncbi.nlm.nih.gov/geo/) under accessions GSE300046 (Repliseq) and GSE299959 (Bru-seq). The source data and code for the main figures can also be found at Zenodo (https://zenodo.org) (https://doi.org/10.5281/zenodo.15678082).

The source data of this paper are collected in the following database record: biostudies:S-SCDT-10_1038-S44318-025-00501-5.

## Peer review information

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

## Acknowledgements

The authors thank B Washburn and S Miller from the Florida State University Department of Biological Sciences Molecular Core Facility for helpful advice, cloning help, and Sanger sequencing services. We also thank Cindy Vied and Yanming Yang from the Florida State University College of Medicine Translational Science Lab for high-throughput sequencing services. This work was supported by NIH grants F31-AG066481 to JLT, R01-GM083337 to DMG and R35-GM128938 to FA. LHG was supported by an NIH training grant to UCSD T32-GM139790.

## Author contributions

**Jesse L Turner**: Conceptualization; Data curation; Formal analysis; Funding acquisition; Validation; Investigation; Visualization; Methodology; Writing—original draft; Writing—review and editing. **Laura Hinojosa-Gonzalez**: Data curation; Software; Formal analysis; Investigation; Visualization; Methodology; Writing—original draft; Writing—review and editing. **Takayo Sasaki**: Resources; Data curation; Supervision; Project administration. **Satoshi Uchino**: Data curation. **Athanasios Vouzas**: Data curation. **Mariella S Soto**: Data curation. **Abhijit Chakraborty**: Software; Formal analysis; Investigation; Visualization. **Karen E Alexander**: Data curation. **Cheryl A Fitch**: Data curation. **Amber N Brown**: Data curation. **Ferhat Ay**: Resources; Formal analysis; Supervision; Funding acquisition; Project administration; Writing—review and editing. **David M Gilbert**: Conceptualization; Resources; Formal analysis; Supervision; Funding acquisition; Methodology; Writing—original draft; Project administration; Writing—review and editing.

Source data underlying figure panels in this paper may have individual authorship assigned. Where available, figure panel/source data authorship is listed in the following database record: biostudies:S-SCDT-10_1038-S44318-025-00501-5.

## Disclosure and competing interests statement

The authors declare no competing interests.

# Expanded View Figures

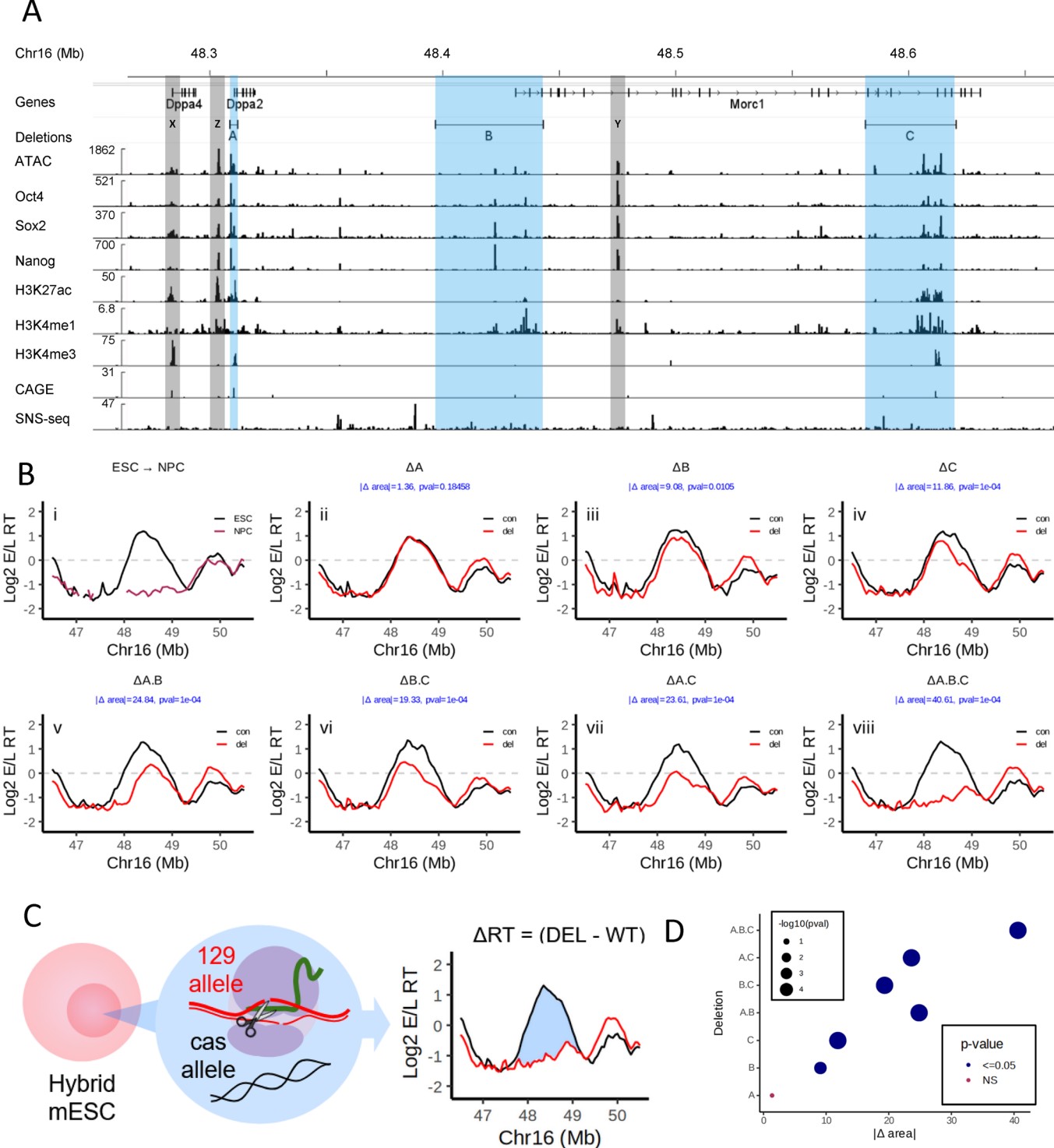

◀ **Figure EV1. Epigenetic signature of the Dppa2/4 replication domain ERCEs.**

(A) IGV browser tracks showing chromatin features of the Dppa2/4 domain (see "Methods" for sources). ERCEs, defined by previous deletions (Sima et al, 2019), are highlighted in blue. X and Y (gray highlights) represent regions that display epigenetic features of ERCEs but did not display ERCE activity in deletion analyses (Sima et al, 2019). (B) Log2(E/L) Repli-seq (Marchal et al, 2018) from (Sima et al, 2019). The first panel (i) shows RT before and after mESC (black) to neural precursor cell (NPCs; maroon) differentiation. The remaining panels (ii–viii) show RT profiles from averaged replicates of independent CRISPR clones harboring deletions of one or more ERCEs on one allele (DEL; red) vs. the homologous unmodified allele (CON; black). Individual replicate experiments are shown in Fig. EV3, and the approach to assess the significance of differences in RT between a deletion allele and the corresponding WT allele are shown in Fig. EV2. (C) Schematic representation of hybrid mESC model, where one allele contains a deletion and the other serves as control, allowing us to study changes in RT by estimating the area under the ΔRT curve in the DPPA domain (see "Methods"). (D) |Δ area| between the averaged RT profile curves of at least two replicates (shown in (C)) and the significance of their delay in RT compared to the corresponding WT alleles. Empirical P values are shown.

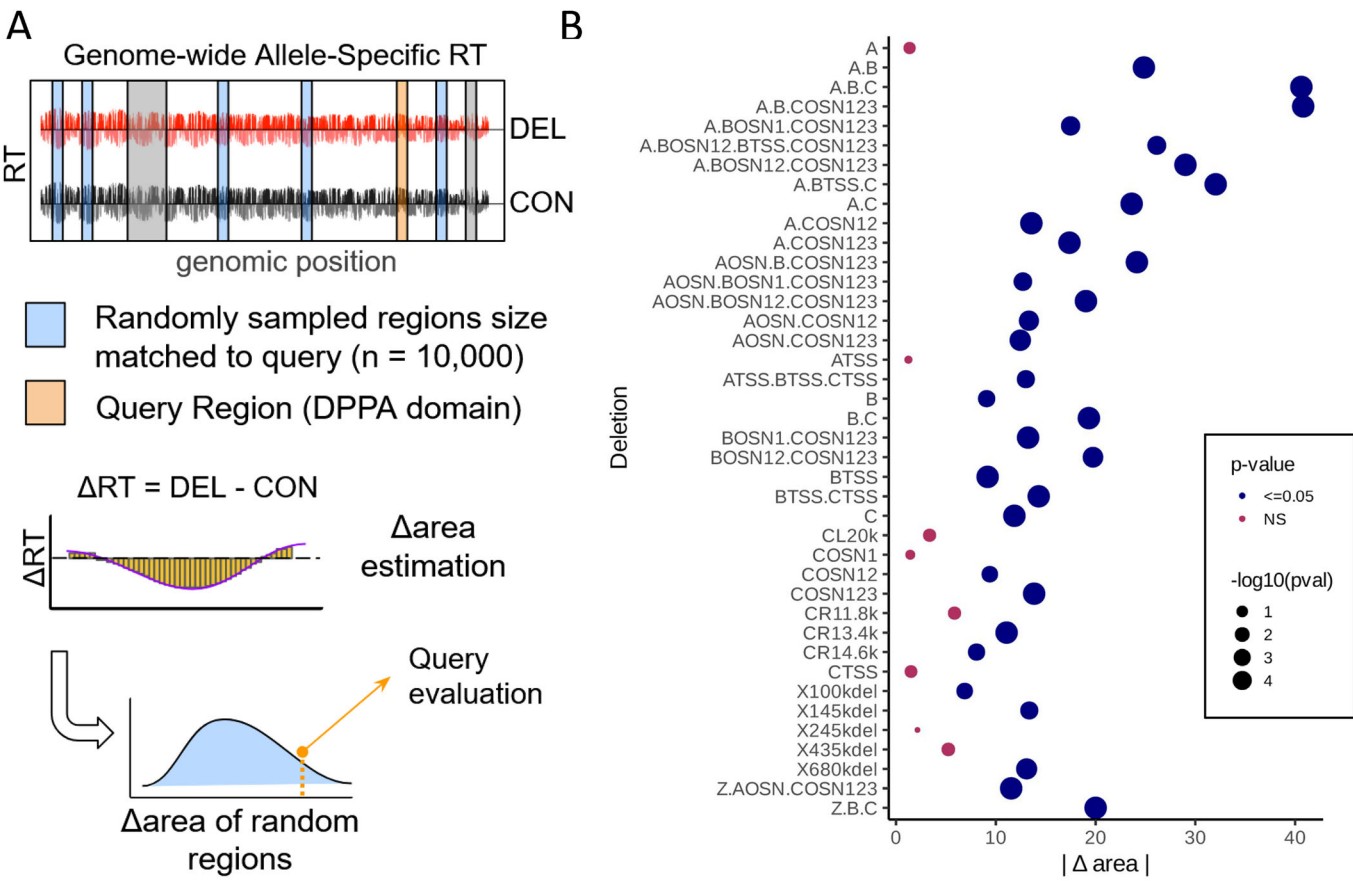

**Figure EV2. Statistical analysis of E/L Repli-seq.**

(A) Schematic representation of AUC approach. (B) |Δ area| for all deletions in this study with corresponding *P* values. Color represents significance at *P* value of 0.05 and size represents −log10(*P* value). Empirical *P* values are shown.

A

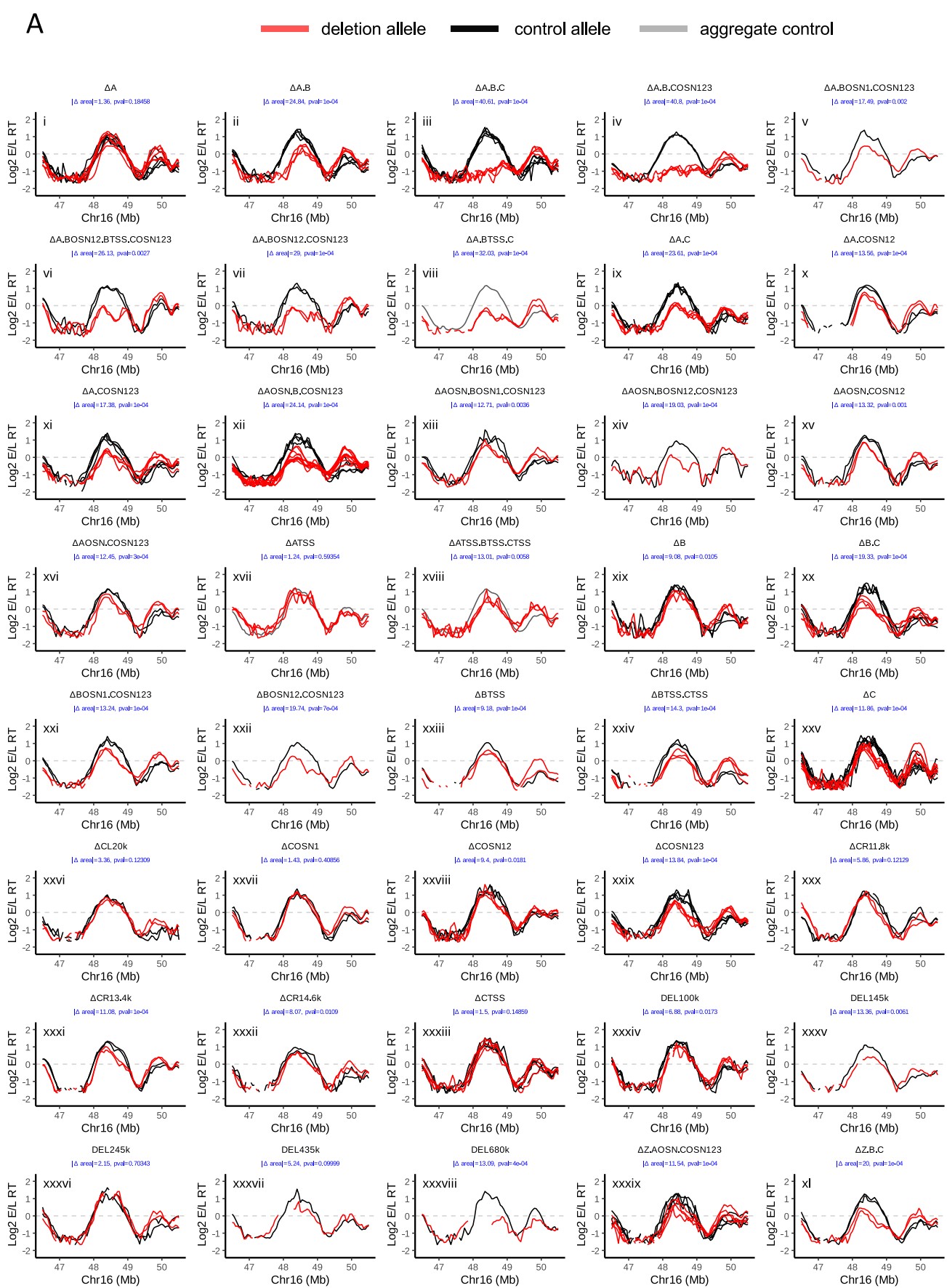

◄ **Figure EV3.  All Repli-seq datasets analyzed in this study.**

**(A) (i–xl)** Allele-specific E/L Repli-seq for all the data generated and analyzed in this study, including relevant deletions from our previous publication (Sima et al, 2019). Replicates (independent CRISPR-mediated deletion clones) are shown as different lines in each plot, comparing the deletion (red) allele to the homologous unmodified allele (black). In the occasional cases where a clone suffered one of the multiple deletions in both alleles we use an aggregated WT control (gray). The aggregated control was generated by averaging all RT profiles from the WT alleles across different deletions. The replication domain to the right of Dppa (c16: 49.65–50.05 Mb mm10) shows some clonal variation in RT but there is a poor correlation ($R = -0.35$) between the effects of the deletions in the Dppa domain to this variation. Rather, this domain has been shown to display a genetic difference in RT between *m. castaneus* and *m. musculus* (Rivera-Mulia et al, 2018) as well as high cell to cell variability in RT (Zhao et al, 2020).

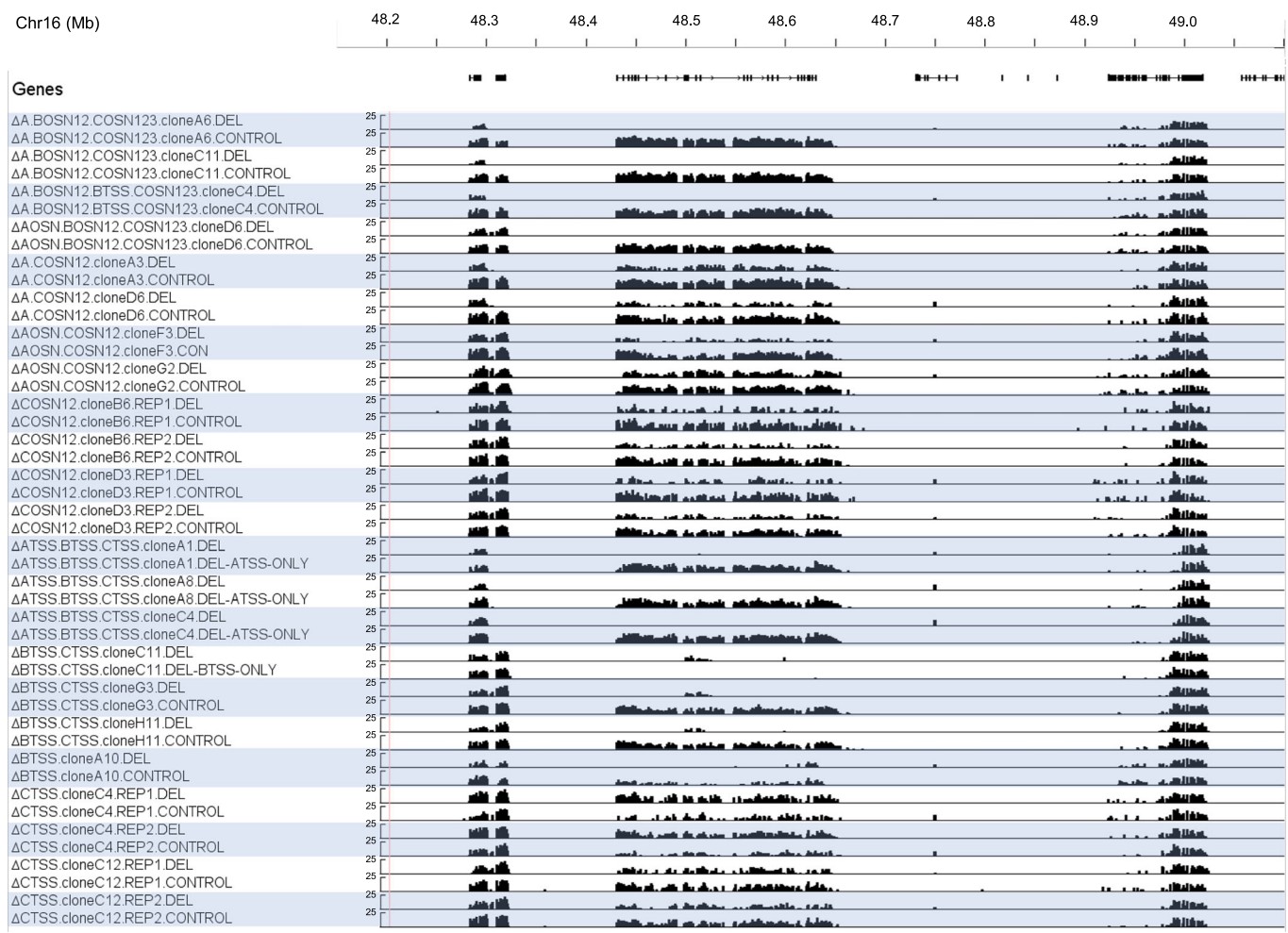

**Figure EV4. All Bru-seq datasets generated in this study.**

All of the BrU-seq generated in this study, displayed as in Fig. 4. OSN deletion series data tracks are found on the top half while TSS deletion series data tracks can be found on the bottom half. We observed some variation in gene expression between the two alleles in WT cells (Fig. 4). This variation can also be seen in the CTSS deletions (bottom tracks), as clone (C4) harbors a deletion of the *castaneus* allele whereas the other clone (C12) harbors a deletion on the *musculus* allele, and neither CTSS deletion has an effect on Morc1 expression.

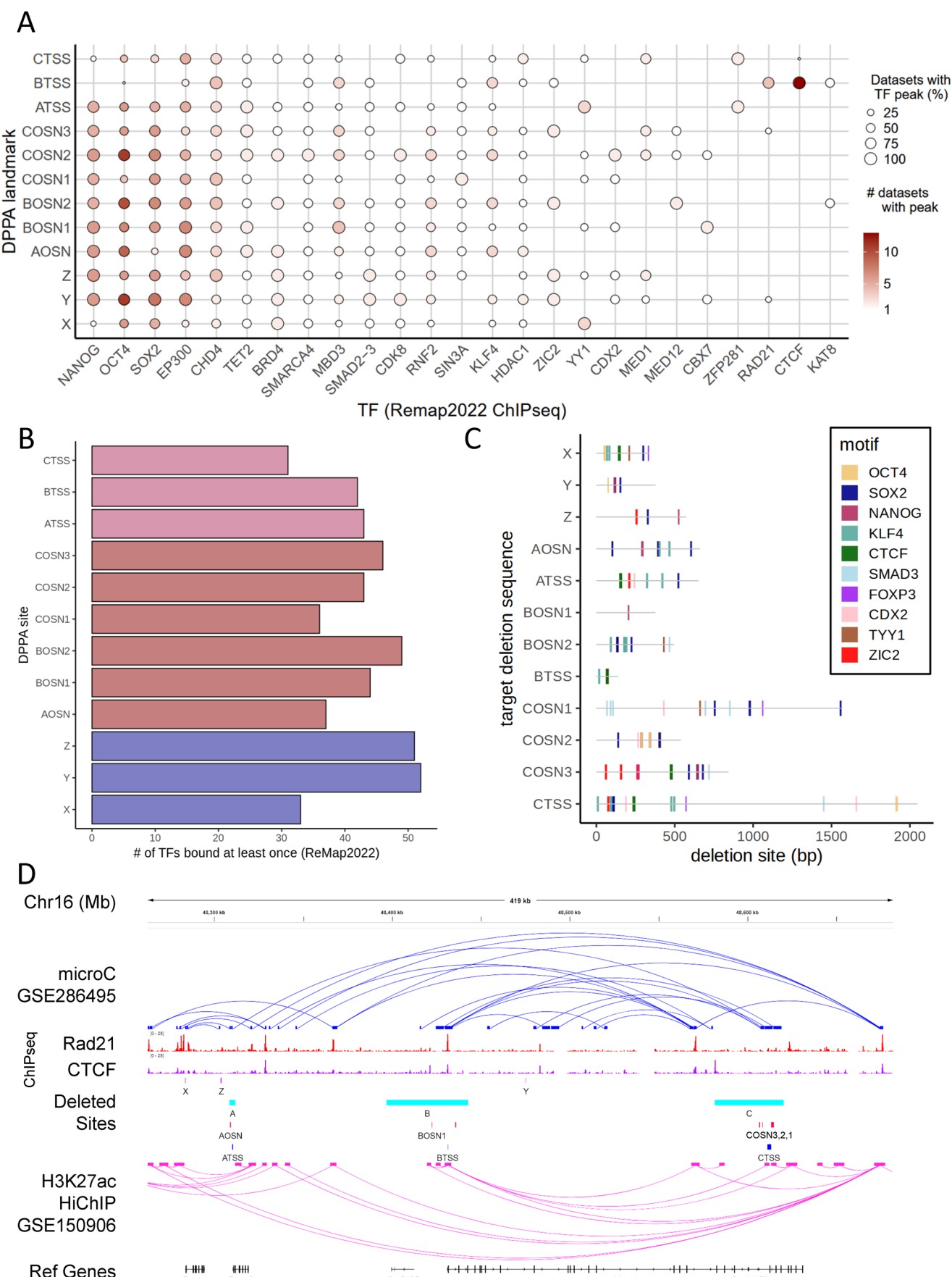

**Figure EV5.  Deleted sites are diverse TFBS.**

(A) ChIP-seq peaks from 'mESC' datasets in Remap2022 overlapping at least two of the sites of interest in the DPPA domain. The size of the dots indicates the percentage of available ChIP datasets with a peak directly overlapping the site, and the color indicates the number of datasets with peak. (B) TF-binding diversity (Hammal et al, 2022) quantified as the number of TFs with at least one ChIP-seq peak from Remap2022 'mESC' overlapping the sites of interest. (C) Putative TFBS in target deletion sequences determined by motif presence. Due to the presence of over 400 TF-binding motifs in this collection of elements, only motifs for TFs from the ChIP-seq targets shown on (A) are included. (D) Micro-C (preprint: Jusuf et al, 2025) and H3K27ac HiChIP loops (Kraft et al, 2022) in the Dppa domain, as well as CTCF and Rad21 ChIP-seq signal over input (Cattoglio et al, 2019; Hansen et al, 2017). Loop resolution varies between datasets and is indicated on *y* axis.

