## [Peer Review File · The EMBO Journal]

Master transcription-factor binding sites constitute the core of early replication control elements

Jesse Turner, Laura Hinojosa-Gonzalez, Takayo Sasaki, Satoshi Uchino, Athanasios Vouzas, Mariella Soto, Abhijit Chakraborty, Karen Alexander, Cheryl Fitch, Amber Brown, Ferhat Ay, and David Gilbert

Corresponding author(s): David Gilbert (gilbert@sdbri.org) , Ferhat Ay (ferhatay@lji.org)

Review Timeline:

Submission Date:	7th Nov 23
Editorial Decision:	19th Dec 23
Appeal Received:	29th Jan 25
Editorial Decision:	26th Mar 25
Revision Received:	17th Jun 25
Accepted:	23rd Jun 25

Editor: Hartmut Vodermaier

Transaction Report:

Prof. David M Gilbert
San Diego Biomedical Research Institute

19th Dec 2023

Re: EMBOJ-2023-116114
Master transcription factor binding sites are required for early replication control element activity

Dear Dr. Gilbert,

Thank you for submitting your manuscript on OSN binding sites and early replication control elements to The EMBO Journal. With some delay due to slow referee responses at this time of the year, we have now obtained the below-copied comments by three expert reviewers. In light of these reports, we unfortunately had to conclude that the study is presently not a strong enough candidate for EMBO Journal publication. As you will see, the referees acknowledge the importance of the topic and appreciate the presented ERCE fine-mapping, as well as the potential interest of the conclusion that master transcription factor binding is key for early replication timing independent of transcription itself. At the same time, they are however concerned that the study remains currently rather descriptive and its scope still too limited, with several immediate mechanistic conjectures (such as epigenome alterations or chromatin accessibility changes) not further analyzed, and key conclusions such as transcription-independence not decisively supported by the present data. I will not go through all their individual points of criticism in detail here, but in light of these combined reservations, I am afraid we find the study still somewhat too preliminary to invite (and thus to some degree commit to) a revised version for consideration as a full-length EMBO Journal article at this point.

That said, should further work along the lines suggested by all three referees allow you to significantly strengthen and deepen the main conclusions, I would not exclude the possibility of once more looking into the study and considering a potential resubmission. In this case, I would however appreciate if you first contacted me with a detailed point-by-point response and summary of new results and their conclusions, so that we could discuss whether a new version might be sufficiently promising to go back to our original referees.

I am sorry that the reports do not allow me to be more positive at this point, but hope that you will nevertheless find the referees' comments and suggestions helpful when considering how to proceed further with this study. Thank you once more for having had the opportunity to consider this work for publication.

Yours sincerely,

Hartmut Vodermaier

Referee #1:

The Turner et al paper seeks to refine the identification of cis-elements involved in the establishment of the Dppa pluripotency-specific early replicating domain in mESCs. A prior publication from 2019 pinpointed ERCEs (Early Replicating Control Elements) situated in three distinct regions named ERCE A, B and C, totaling approximately 80 kb. Based on genome-wide ChIP data, the authors previously suggested that the primary mediators of this regulation were promoters and enhancers containing Oct4/Sox2 and Nanog co-bound with CBP/p300. In this study, the authors performed a remarkable series of more precise deletions, compellingly demonstrating the involvement of co-bound OSN (Oct4/Sox2/Nanog) sites in early replication control. However, the authors' demonstration of the claim that transcription is unnecessary for early RT of the Dppa domain is inadequate. Additionally, the statement that "SNS origins and the TSS are dispensable for ERCE activity" is incorrect based on the provided information. The data implies a more intricate relationship between transcription factors, TSS (and potentially transcription), and origins in establishing the ESC-specific early domain. This finding concurs with existing studies on how constitutive early replication domains are formed. However, this new research lacks context within the literature. The data discovered at this specific early domain in mESCs appears to align with the already described mechanisms. Hence, commenting on the similarities rather than attempting to demonstrate different regulation would be of potential interest. It would greatly benefit the community to have a comprehensive and accurate understanding of this regulation on a global scale rather than a simplified

and incorrect version.

Major concerns

Study of ERCE C

1) The precise size of deletions is impossible to determine. By examining the figures, I found that deletions might range from 0.5 to 2 kb. The authors need to provide precise coordinates of the deletions in the material and method section for the reader to accurately identify the removed potential cis-regulatory elements. A table of guide RNAs used to make deletions should also be added together with the real deletion observed after sequencing of PCR fragments.

2) The authors start with the dissection of the 39 kb ERCE C. They describe four elements, 3 OSN (COSN1-3) and one intragenic TSS named CTSS. As this site is also bound by Oct4/Sox2/Nanog (Figure 2A), like the TSS found in the ERCE A, it is difficult to figure out why the latest element is named ATSS/AOSN2. In order to be consistent, CTSS should be called CTSS/COSN4.

3) According to the authors, "deletion of the right part of ERCE C (Δ CRight14.6k) causes a delay similar to the original C deletion". It is obvious that it is not the case, the deletion gives a milder effect. The authors should be more precise on the interpretations and conclusions. There is therefore something in the left part that contribute to the effect.

Later, the authors mention that a smaller "deletion (Δ CRight11.8k) results in a partial delay of RT" which to my point of view is similar to the larger deletion (Δ CRight14.6k). Another problem with this deletion is that in most cases two clones are shown and it seems that for this deletion only one clone was analyzed.

Finally, the authors mention "we also generated a larger deletion that spans all 3 OSNs (Δ Crighth13.4k), which also gave a full "C effect" similar to Δ COSN123 (Fig. 2B)." The results of this clone should help the reader to be convinced that there is nothing important in the left part. Unfortunately, RT profiles of the clone are not shown in Figure 2B or elsewhere in the paper.

Altogether, these large deletions are not well described.

4) The authors provide a map of repli-seq (panel A) in Supplementary Figure 1, which should give us information about the sites of replication initiation. The authors should be aware that while it is clear that replication origins must be within the earliest replicated region, it is impossible to draw any conclusions for the rest of the region, as origins may fire sequentially at different times, resulting in an RT shape similar to that shown in Supplementary Figure 1A. The community has agreed that the only methods that allow precise mapping of replication origins are SNS-seq and Ini-seq. These two methods overlap well in early regions, while SNS-seq has the advantage of providing information on origins that fire later in S phase. The authors use an SNS-seq dataset generated in M. Mechali's lab to define initiation sites. I would encourage the authors to remove the RT timing data that provide any information on origin positioning and to provide a more readable picture of SNS-seq so that the reader can truly evaluate sites of enriched SNS that are preferential sites of initiation. Furthermore, the authors show data from Prorok et al, 2019, which is fine, but to strongly support their model, they can also use another good dataset that was recently published (Jodkowska et al, 2022). Three strong initiation sites are observed in most conditions used in this paper. The strongest sites are located outside the deleted regions. These strong origins could be co-operating with the OSNs found within the regions. I encourage the authors to take into account this new and robust study. For all these reasons, I think that the statement "the left half of C contains the only replication origin activity detected by SNS-seq" is incorrect based on the recent and available SNS data, since strong sites are mostly outside deleted regions.

5) I do not understand the sentence "These observations demonstrate that ERCEs regulate RT independently of origin choice".

6) The results described for smaller deletions are convincing, Δ CONS123 and Δ ABCONS123 have profiles similar to Δ C and Δ ABC respectively. It is therefore difficult to understand why Δ CONS123 has more effect than the full (Δ CRight14.6k). The authors should explain this point and also show their results with (Δ Crighth13.4k).

Transcription is not necessary for early RT of the Dppa Domain

7) In the last part of the results, the authors sequentially delete promoters named A, B and C, which control the expression of the Dppa2 and Morc1 genes. As they correctly mention, Δ ATSS and Δ BTSSCTSS result in a near-complete loss of Bru-seq signal so it is not a total loss. It is also difficult to accurately assess the magnitude of the fold change as the signal on the Myh15 gene, which should remain the same in each sample, is the strongest in the two samples where transcription is not affected (Figure 4A, samples Control and Δ CTSS). This is also the case for Dppa4. A precise quantitative study with RT-qPCR at different positions along the Dppa2 and Morc1 genes should be performed. Another problem is that not only is weak transcription observed along the Dppa2 and Morc1 genes, but it is also accompanied by the appearance of pervasive transcription outside the body of the genes. In addition, the Δ ATSSBTSSCTSS induces a weak but significant RT delay. In summary, this mutant not only shows weak transcription along the entire locus, but also a small shift towards later replication. Therefore, the authors' conclusion "that transcription is not necessary for early RT at the Dppa domain" is not convincingly demonstrated.

8) Of the three independent TSS deletions, only the B gives a modest RT delay. So the authors have made a deletion in a context where ERCE A and C are already removed (Δ ABTSSC) and this further increases the RT delay, but less than the full Δ ABC. The authors are trying to make a suggestion that is not entirely satisfactory.

For all these reasons, the demonstration that "transcription is not necessary for early RT of the Dppa Domain" is not well supported. How strong sites of replication initiation are involved is also not well described in this form of the paper. It seems more plausible that a complex cooperation between all these elements is important for the efficient establishment and maintenance of the domain. This model has been proposed in several studies attempting to address this specific question, and I think that the paper would be greatly improved if the authors put previous data into perspective.

Minor comments

1) Figure 1 shows only the data already published in Sima et al, 2019. Although it is important to have this figure as a reference (note that Δ C, Δ AC and Δ BC are also shown in Figures 2, 3 and 4 to facilitate the comparison of the profiles), it could be

included as a supplementary figure and only discussed mainly in the introduction.

2) As mentioned above, there is no visible difference in RT within the Dppa domain when a large deletion of the left part of ERCE C (Δ CLeft20k) is made. However, one can observe a difference in the RT of a small domain located at position 50 on Chr16. Surprisingly, the RT of the modified allele is more advanced (Figure 2B). This is also the case for several mutants (Δ COSN123, figure 2B...). Can the authors give us an interpretation of these results?

3) The resolution and quality of the figures are really poor and makes it even more difficult to follow the story.

Study of ERCE A and B

4) This part starts with deletions within the ERCE A. To make it easier to understand, I would put the RT profiles being compared on the same line and in the order in which they appear in the text. So on a first line I would put Δ AC (Not really necessary); Δ AOSN1COSN123 and Δ ACOSN123 together.

5) There is only one clone described for Δ BOSN12COSN123.

Referee #2:

The manuscript "Master transcription factor binding sites are necessary for early replication control element activity" by Turner et al detailed the cis elements, early replication control elements (ERCEs), that is necessary for early replication in mESCs. They Used CRISPR-Cas9 to map the sites of 3 ERCEs within the Dppa domain. They deleted putative cis elements on one allele while the other allele can serve an internal control in mESCs derived from hybrid mice (F121-9). They found that one or more sites for co-binding of the pluripotency transcription factors Oct4, Sox2, Nanog (OSN) are required for early replication, whereas TSSs are not. So they propose that these master pluripotent transcription factor binding sites (OSN sites) are important for early replication through a transcription-independent way. Overall, the study is well-designed and represents a significant progress from their 2019 original work. With this said, I have the following comments.

Major concerns

1. OSN sites mapped here might still be too large to exclude elements other than OSN sites. For example, are there FoxP binding sites in OSN sites? One of the doable convincing strategies is to artificially recruit OSN TFs, or one of them or DBF4 as discussed in the next points, in the cis-deleted background like Δ ABCOSN123, to see whether OSN sites are passable or not. If it works, the DNA-binding mutants of these TFs can be used as an ideal control. All these will also amend the complete missing of the trans factor part in current stage.
2. Since DDK defines the first kinase to activate MCM helicase during S phase. Have authors checked their enrichment by DDK-ChIP? This may also help to address why OSN (Z) upstream AOSN, X and Y have no ERCE activity.
3. The authors deleted the whole set of all three OSN sites, and conclude a co-binding model. Did authors try to further delete all binding sites for one of these TFs within ERCEs?
4. Z represents an OSN site near A that, like X and Y shown in Fig.1, does not harbor ERCE activity
5. In Discussion part, authors proposed that OSN TFs or other OSN-interacting TFs might be required. Did they try to deplete them or one of them (as in the co-binding model) to test it?
6. In Discussion part, an alternative possibility is that OSN sites directly mediate 3D chromatin interactions, which also explains transcription-independent, or even OSN TFs-independent.

Minor comments:

1. It would be better to describe how RT is analyzed statistically in details to conclude a significant change/delay or not, for example Δ C, Δ CLeft20k and Δ CRight14.6k in Fig 2B.
2. The last sentence of Abstract, "pluripotency transcription factor binding sites ensure early replication independent of transcription, suggesting a means for co-regulation of RT with cell fate transitions during development." I cannot clearly understand the logic connection between "independent of transcription" and co-regulation. Pls clarify it in Discussion.

Referee #3:

The authors conducted deletion analyses of the ERCEs A, B and C, with a particular focus on Oct4, Sox2, Nanog (OSN) binding sites. The two major conclusions from this study are 1) OSN binding sites present in these sequences are required for early replication control. 2) Early replication does not require active transcription. The replication timing data presented are convincing and are consistent with the conclusions.

1 Unfortunately, the study fails to provide precise mechanistic insight into the early replication control by the OSN binding. Although several possibilities can be envisioned, the experiments to test the models have not been performed.

For examples, authors eluded the potential similarity between the actions of pluripotent transcription factors and budding yeast

Fkh1/Fkh2. This could be a good analogy, which can be experimentally tested.

2 Obvious questions arising from the data presented in the manuscript have not been experimentally addressed.

2-1 What is the epigenetic state of the OSN mutants? Related to this question, what is the correlation between the superenhancer and what the authors call "replication enhancer"?

2-2 What is the chromatin openness of the mutant cell lines? Does open chromatin induced by the OSN binding explain the early replication?

3 Data show that deletion of TSS abrogates transcription but does not significantly affect early replication, showing that the transcription is not required for early replication. How about the transcription in these OSN binding site mutants? Is it affected by the OSN binding sites mutant cells[;x/]?

4 Other questions:

4-1 This study shows that OSN binding sites are required for early replication, but it is not clear if they are sufficient. Can authors reconstruct the early replication by combining the OSN binding sequences?

4-2 X,Y and Z represent the OSN that are not required for ERCE. What are the differences between these and those OSN binding sites in ERCE. Can the former replace the latter? Or are there intrinsic differences between these OSN binding sites?

Other comments:

Deletion endpoints (relative to the OSN binding sites) for each mutant cell line need to be precisely described.

General suggestion on the nomenclature of the mutant cell lines:

I think it would be easier for the readers to understand each deletion cell lines if nomenclatures are changed a bit.

All the cell lines are combinations of different deletions. Each deletion could be preceded by Δ . (e.g. Δ AOSN1 Δ COSN123 rather than Δ AOSN1COSN123; Δ A Δ B Δ COSN123 rather than Δ ABCOSN123). This would make it easier for the readers to grasp the combination of the mutations from each domain.

In Figure 3C

Δ AOSNBCOSN123 should be Δ AOSN1BCOSN123 (Δ AOSN1 Δ B Δ COSN123)

ZAOSNCOSN123 should be Δ ZAOSN1COSN123 (Δ Z Δ AOSN1 Δ COSN123)

ZBC should be Δ ZBC (Δ Z Δ B Δ C)

Figure 1 is the summary of the previous publications, and can be presented as a Supplementary Figure.

*** As a service to authors, The EMBO Journal offers the possibility to directly transfer declined manuscripts to another EMBO Press title (EMBO Reports, EMBO Molecular Medicine, Molecular Systems Biology) or to the open access journal Life Science Alliance launched in partnership between EMBO Press, Rockefeller University Press and Cold Spring Harbor Laboratory Press. The full manuscript (including reviewer comments, where applicable and if chosen) will be automatically forwarded to the receiving journal, to allow for fast handling and a prompt decision on your manuscript. For more details of this service, and to transfer your manuscript to another EMBO title please follow this link:

Link Not Available

David M. Gilbert, Ph.D.
Professor
3525 John Hopkins Court
San Diego, CA 92121
Email: gilbert@sdbri.org

February 10, 2025

RE: EMBOJ-2023-116114R

Dear Dr. Vodermaier,

Please find our revised manuscript “**Master transcription factor binding sites constitute the core of early replication control elements**”.

First, we want to apologize for the very long delay in returning this manuscript. In addition to making new cell lines and carrying out new experiments and analyses, the first author is now in his post-doctoral laboratory with limited time to participate. He also suffered a personal family tragedy during this time. We hope that the original reviewers will still be available, as we have put a lot of effort into answering every one of their criticisms.

We want to thank the reviewers for such thoughtful criticisms of our manuscript. We spent a great deal of time in Zoom meetings discussing these points and the manuscript has benefitted greatly. In summary, here is a bullet list of what we have done in revising the manuscript:

New Data

- Engineered one additional cell line (two technical replicates) containing $\Delta A.BOSN12.BTSS.COSN123$ to investigate the effects of $\Delta BTSS$ on $\Delta A.BOSN12.COSN123$. New data for this clone includes multiple replicates of both Repli-seq and Bru-seq. This brings the total number of newly engineered cell lines in this manuscript to **57**.
- Performed **23** new BrU-seq datasets, including **6** new experiments in clones not previously analyzed and all **17** prior datasets re-sequenced with identical chemistry to assist with background assessment.
- Performed **12** new Repliseq experiments, addressing the new clones and also generating more replicates for deletions that gave subtle RT effects and whose quantitative effects on RT were in question by reviewers.
- ***In total, this study comprises 232 RepliSeq and 46 BrU-seq allele-specific datasets in F121-9 hybrid mESCs, encompassing 79 genetically distinct clones (including 22 that were previously published and re-analyzed).***

New Analysis

- Established a quantitative framework to measure and statistically assess the effect of each deletion on RT, as well as to compare deletions to each other. These statistics are now included in Figures S1, 1,2,3 and 4 and Table 4.
- Re-analyzed all Bru-seq data with a new quantification method (Figure 4, Supp Table 5)

- Expanded our understanding of the deleted sites by leveraging publicly available datasets from:
 - Two additional SNS-seq datasets from Jodkowska et al and Pratto et al (Figure 5)
 - OK-seq signal tracks and IZ peak calls from Petryk et al (Figure 5)
 - ChIPseq peaks from Remap2022 (Figures 1-3, S5)
 - HiChIP loops from the LoopCatalog (Figure S5)
 - Ultra deep Micro-C (from Jusuf et al. 2025, BioRxiv) (Figure S5)
 - CTCF and Rad21 ChIPseq (from Hansen et al. 2017) (Figure S5)
 - TF motif analysis on the deleted sites (Figure S5).

Manuscript Improvements

- Provide a new model Figure 6, as well as 3 additional supplemental figures and 5 new Supplemental Tables.
- Re-made all Figures and re-arranged Figures in response to reviewers' suggestions.
- Employed a new nomenclature for our mutant cell lines to improve readability and consistency
- Re-wrote the entire manuscript, with particular attention to any reference to origins and transcription to address reviewers' critiques.
- Added a proper Discussion section.
- Re-named OSNs as "subERCES" to develop a clearer description of their activities.

We are in the process of uploading all datasets to GEO.

In addition to the reviewer requests, you also requested that we expand the paper and develop a devoted Discussion section, which we have now done, including now a total of 6 main figures, 5 supplemental figures and 5 supplemental Tables.

A few caveats that should be stated up front. Every person who looks at our data immediately thinks of one more deletion that would be useful to have, but each person picks a different one and none of them think of how long it takes to make one cell line. I hope we can get beyond this type of "your favorite experiment to do next" stage; each of these deletions requires months of cell line validation, followed by experimentation and, finally, analysis. Also, deletions give diminishing returns and Crispr is not a scalpel; one cannot do precision deletions due to the need for PAM sequences and efficient gRNAs and avoidance of repetitive and difficult DNA sequences. As asked by one reviewer, we have started the process of inserting these candidate sequences into ectopic sites, which promises to eventually give us a more robust system, but this is an extensive effort by a new post-doc and will take a lot of hard work and careful analysis and should be developed as its own separate story. Our work provides novel insight into an extremely complex large scale chromosome structure and function problem and it provides a huge resource of cell lines that others can use to answer myriad questions we get asked about these cell lines; a potential goldmine for those interested in 3D architecture for

example. This paper represents a tremendous amount of work! We hope that you and the reviewers will enjoy it, and will appreciate their own contributions reflected in this revised version.

On the following pages, we provide you with a point by point response, **in blue font**, of how we intend to respond to the reviewers, with the changes made to the manuscript **in bold blue font**:

Sincerely,

David M. Gilbert, Ph.D.
Professor
San Diego Biomedical Research Institute

Referee #1:

The Turner et al paper seeks to refine the identification of cis-elements involved in the establishment of the Dppa pluripotency-specific early replicating domain in mESCs. A prior publication from 2019 pinpointed ERCEs (Early Replicating Control Elements) situated in three distinct regions named ERCE A, B and C, totaling approximately 80 kb. Based on genome-wide ChIP data, the authors previously suggested that the primary mediators of this regulation were promoters and enhancers containing Oct4/Sox2 and Nanog co-bound with CBP/p300. In this study, the authors performed a remarkable series of more precise deletions, compellingly demonstrating the involvement of co-bound OSN (Oct4/Sox2/Nanog) sites in early replication control. However, the authors' demonstration of the claim that transcription is unnecessary for early RT of the Dppa domain is inadequate. Additionally, the statement that "SNS origins and the TSS are dispensable for ERCE activity" is incorrect based on the provided information. The data implies a more intricate relationship between transcription factors, TSS (and potentially transcription), and origins in establishing the ESC-specific early domain. This finding concurs with existing studies on how constitutive early replication domains are formed. However, this new research lacks context within the literature. The data discovered at this specific early domain in mESCs appears to align with the already described mechanisms. Hence, commenting on the similarities rather than attempting to demonstrate different regulation would be of potential interest. It would greatly benefit the community to have a comprehensive and accurate understanding of this regulation on a global scale rather than a simplified and incorrect version.

This is a complex topic and we agree that we need to do a better job explaining the conclusions. As far as regulation on a global scale, at present, our conclusions are limited to one locus. We intend to convey here that ERCEs have revealed a mechanism for regulating replication timing that is linked to master TFs and cell fate changes. Our original deletions were very large and responsible for several activities; early RT, transcription, compartmentalization and chromatin loop interactions were all affected. The current work aims to tease out the contributions to RT by transcription via the enhancers and promoters contained within the ERCE deletions from non-transcriptional contributions to RT. We did not mean to imply that transcription cannot or does not make a contribution to RT of the Dppa locus. **To underscore this point, we have now designated the elements within ERCEs that have transcription-independent contributions to RT “subERCEs” and distinguish them from elements that affect RT via transcription alone (e.g. BTSS).** subERCEs affect transcription, but they also clearly have roles in RT that do not require transcription as nearly all transcription can be eliminated without eliminating early replication (TSS deletions; ab deletion of Sima, 2019).

As for origins, we now provide additional data supporting a stochastic model for initiation in the Dppa locus, and we do not find any evidence of a role for any specific origin in RT. The data in Figure 5 show that replication timing is regulated at the domain level, and there is a great deal of flexibility as to where early replication can initiate.

As detailed below, we have carefully re-written the manuscript to explain our conclusions more completely and, as suggested by the editor, we have broken out a separate Discussion section that includes sections devoted to addressing each of these two subjects, transcription and origins.

We are not clear on what the reviewer means by “lacks context with the literature”, but we are assuming that it refers to the prior sentence “concur with existing studies on how constitutive early replicating domains are formed”. The reviewer does not cite any studies; perhaps by “constitutive” the reviewer is referring to yeasts? Our model is well grounded in the budding yeast literature (particularly Aparicio) that we cite prominently as a major driver of our model. The fact that developmental control of replication timing in mammals may follow a similar model to paradigms of how constitutive early replication domains are formed in budding yeast is, in our opinion, very exciting. **We have made an effort to make this point even more transparent in this revised manuscript.**

Major concerns

1) The precise size of deletions is impossible to determine. By examining the figures, I found that deletions might range from 0.5 to 2 kb. The authors need to provide precise coordinates of the deletions in the material and method section for the reader to accurately identify the removed potential cis-regulatory elements. A table of guide RNAs used to make deletions should also be added together with the real deletion observed after sequencing of PCR fragments.

We apologize this was not provided in the submitted manuscript. **We now provide three tables (Supplementary Tables 1-3) listing all sgRNA and PCR primer sequences and coordinates, deletion coordinates and sgRNA pairs, and cell line genotypes.**

2) The authors start with the dissection of the 39 kb ERCE C. They describe four elements, 3 OSN (COSN1-3) and one intragenic TSS named CTSS. As this site is also bound by Oct4/Sox2/Nanog (Figure 2A), like the TSS found in the ERCE A, it is difficult to figure out why the latest element is named ATSS/AOSN2. In order to be consistent, CTSS should be called CTSS/COSN4.

Thank you for pushing us on this. We have now quantified the strength of each OSN as the frequency that they appear in the many datasets in the literature from mouse ESCs, curated by a recent publication. By this quantification, the CTSS is a weak OSN site. We agree that it was confusing to call the ATSS an ATSS/AOSN without a quantitative definition. We now call any TSS simply a TSS, even if it has some evidence of OSN binding. **We have now quantified peak calls using a large collection of ChIP-seq datasets from the Remap Database and provide a figure panel for the quantification of the strength of each OSN in each of the three ERCE dissection figures 1-3.**

3) According to the authors, "deletion of the right part of ERCE C (Δ CRight14.6k) causes a delay similar to the original C deletion". It is obvious that it is not the case, the deletion gives a milder effect. The authors should be more precise on the interpretations and conclusions. There is therefore something in the left part that contribute to the effect.

Thank you again for pushing us to clarify these complicated comparisons. In the prior version of the manuscript, we were relying on visual inspection of the data. With the very large differences in RT that we reported in 2019, this was reasonable. Here, we are trying to compare more subtle differences in RT and the reviewers criticism underscores the need for quantitative analysis. Taking advantage of our hybrid mouse system, we have measured the area between WT and mutant allele RT profiles and developed a statistical algorithm to assign significance to those differences. We also performed additional replicate RT of several clones to confirm reproducibility. The results show that all deletions that remove all three COSNs are

not statistically different from each other or from the full C deletion. **We now provide Δ area (area between WT and mutant alleles), and p-value of the difference between WT vs. mutant RT in each graphical panel. We also provide figure panels that co-plot key deletions on the same graph as subtractions from WT. We provide figure panels illustrating the comparative Δ area and p-values for all cell lines in each figure. We also provide Supplemental Table 4, which shows the statistical differences between each deletion and WT, as well as pair-wise comparisons of key deletions in the manuscript, respectively.**

Later, the authors mention that a smaller "deletion (Δ CRight11.8k) results in a partial delay of RT" which to my point of view is similar to the larger deletion (Δ CRight14.6k). Another problem with this deletion is that in most cases two clones are shown and it seems that for this deletion only one clone was analyzed.

For the significance of each RT delay, we hope the new statistical analysis described above, including the pair-wise statistical comparisons in Supplemental Table 4, will be satisfactory. As to the number of clones for each deletion series, although we strived for two clones of each deletion, we did not succeed in every case. Δ CRight11.8k was one of those cases. **To mitigate the lack of multiple independent Crispr clones, we have now performed multiple experimental replicates of Δ CRight11.8k to increase confidence in the RT profiles.**

Finally, the authors mention "we also generated a larger deletion that spans all 3 OSNs (Δ Crigh13.4k), which also gave a full "C effect" similar to Δ COSN123 (Fig. 2B)." The results of this clone should help the reader to be convinced that there is nothing important in the left part. Unfortunately, RT profiles of the clone are not shown in Figure 2B or elsewhere in the paper. Altogether, these large deletions are not well described.

This was another oversight on our part. We somehow left this data out of the figure. We now provide this data in the revised manuscript.

4) The authors provide a map of repli-seq (panel A) in Supplementary Figure 1, which should give us information about the sites of replication initiation. The authors should be aware that while it is clear that replication origins must be within the earliest replicated region, it is impossible to draw any conclusions for the rest of the region, as origins may fire sequentially at different times, resulting in an RT shape similar to that shown in Supplementary Figure 1A. The community has agreed that the only methods that allow precise mapping of replication origins are SNS-seq and Ini-seq. These two methods overlap well in early regions, while SNS-seq has the advantage of providing information on origins that fire later in S phase. The authors use an SNS-seq dataset generated in M. Mechali's lab to define initiation sites. I would encourage the authors to remove the RT timing data that provide any information on origin positioning and to provide a more readable picture of SNS-seq so that the reader can truly evaluate sites of enriched SNS that are preferential sites of initiation. Furthermore, the authors show data from Prorok et al, 2019, which is fine, but to strongly support their model, they can also use another good dataset that was recently published (Jodkowska et al, 2022). Three strong initiation sites are observed in most conditions used in this paper. The strongest sites are located outside the deleted regions. These strong origins could be co-operating with the OSNs found within the regions. I encourage the authors to take into account this new and robust study. For all these reasons, I think that the statement "the left half of C contains the

only replication origin activity detected by SNS-seq" is incorrect based on the recent and available SNS data, since strong sites are mostly outside deleted regions.

ERCEs "talking" to origins has been an integral part of our working model. We do not know whether these are direct interactions, or whether the ERCEs create a phase separated droplet or "micro-environment". Given the highly stochastic nature of mammalian replication origins, and the particularly broad origin usage at the Dppa locus, we do favor a droplet or micro-environment model. **We now make this point more prominent in the Discussion and provide a model Figure 6.**

We apologize that we did not show all the mESC replication origin datasets in the literature. We showed the Prorok et al, 2019 data because Marcel shared that data with us prior to publication and it has been in our posters and slide presentations for a long time. We should have shown all origin mapping data that has come out since our 2019 ERCE paper. We appreciate the reviewer's point. We were hoping to avoid triggering the 40+ year long debate about the nature of mammalian origins, dating back to the very early 1980s experiments of Ron Laskey, Joyce Hamlin and Michelle Calos; a debate that still rages and devolves into technical details of lambda exonuclease, G4 quadruplexes and drug artifacts. The important point is that, we hope, the reviewer agrees that ERCEs are not origins. They have origins nearby, they may even experience initiation on rare occasions not detected by SNS-seq, but they are not devoted sequences for initiation; initiation can take place at many sites within the domain.

We agree that SNS-seq and Ini-seq are the only methods that map initiation sites within a kilobase, but not without caveats; Ini-seq can detect only a fraction of origins in synchronized cells, both are ensemble methods and can only detect efficient origins. For example, SNS detects very few origins in late regions where there must be many inefficient sites of initiation in order to replicate large heterochromatic regions in short periods of time. Ini-seq uses G1/S synchronization and detects < 15% of SNS-seq origins. Regardless, my lab has always tried to remain balanced and objective on this issue and not ignore any data in the literature; usually the data are fine when properly interpreted. Single molecule methods (ORM and ONT) have shown quantitatively and indisputably (albeit at 15kb resolution) that 80% of initiations are outside of the initiation zones and fire at an undetectably low frequency. Only 20% of initiations occur within initiation zones of ave. 45kb that are used 5-20% of cell cycles. Within those zones there are many sites that can be used at variable frequencies. **To put these findings in the context of ERCE activity, and to do justice to the reviewers' questions, we now have added this information to a Discussion section that focuses on origins.** Since each method has its strengths and weaknesses with respect to resolution, stringency and accuracy, we submit that it is best to take into account all available data types. **Thus, in revised Figure 5, we now provide all data in mESCs published since 2019, including Jodkowska et. al.** We thank the reviewer for encouraging us to collect all of this data as it adds a lot to the picture of how the Dppa replication domain replicates. It is clear that both Ok-seq and Repli-seq identify the initiation zone covering all three ERCEs and that the 245k deletion that we previously showed in Supplemental data deletes the 5 strongest 50kb windows of earliest DNA synthesis, the Ok-seq polarity transition and all Jodkowska SNS-seq peak calls with no effect on RT. It also shows that the SNS-seq peak calls from any of the published studies do not overlap subERCEs.

5) I do not understand the sentence "These observations demonstrate that ERCEs regulate RT independently of origin choice".

We deleted almost all the strong SNS origins and there was no change in RT. Since replication is still occurring early in S, the relative efficiencies of origins necessarily have been re-distributed, but RT still depends upon the ERCEs; thus, ERCEs are agnostic to origin choice. **We have eliminated the phrase "origin choice" from the manuscript and state simply that RT is independent of any specific initiation sites.**

6) The results described for smaller deletions are convincing, Δ CONS123 and Δ ABCONS123 have profiles similar to Δ C and Δ ABC respectively. It is therefore difficult to understand why Δ CONS123 has more effect than the full (Δ CRight14.6k). The authors should explain this point and also show their results with (Δ Crighth13.4k).

Please see our response to criticism #3 above. In brief, statistical analyses show that all deletions that include all three COSNs are not significantly different from each other.

Transcription is not necessary for early RT of the Dppa Domain

7) In the last part of the results, the authors sequentially delete promoters named A, B and C, which control the expression of the Dppa2 and Morc1 genes. As they correctly mention, Δ ATSS and Δ BTSSCTSS result in a near-complete loss of Bru-seq signal so it is not a total loss. It is also difficult to accurately assess the magnitude of the fold change as the signal on the Myh15 gene, which should remain the same in each sample, is the strongest in the two samples where transcription is not affected (Figure 4A, samples Control and Δ CTSS).

This is also the case for Dppa4.

The appearance of increased transcription outside of the affected regions is due to a typical genomics problem when datasets of different Illumina chemistries (in this case 50PE vs. 150SE) are integrated computationally. To directly address the reviewer's concern, **we have re-sequenced all of the libraries using the same chemistry and repeated the analysis.** The positive control genes in the neighboring domain are now equivalent in all experiments, and normalizing data collected with similar chemistries has also reduced the noise in the datasets.

A precise quantitative study with RT-qPCR at different positions along the Dppa2 and Morc1 genes should be performed.

We disagree with this approach. We have consistently found Bru-seq (and deep genomics in general) to be more quantitative than single locus RT-qPCR results. Bru-seq provides datasets where each bin is a replicate of an adjacent bin within the body of each gene. It is consistently more reliable in our nearly 20 years of genomics (array and seq) vs. PCR experience. We agree that the data using different sequencing chemistries (done to save money and time by getting into sequencing lanes with other labs) was creating unnecessary noise. We believe that the data re-sequenced with matching Illumina chemistry, is now convincing.

Another problem is that not only is weak transcription observed along the Dppa2 and Morc1 genes, but it is also accompanied by the appearance of pervasive transcription outside the body of the genes.

This is also an artifact of the same Illumina chemistry problem, as discussed above. This non-genic noise is not present in the new datasets.

In addition, the Δ ATSSBTSSCTSS induces a weak but significant RT delay. There are many possible explanations for this

The point is that a substantial amount of transcription can be eliminated while remaining early replicating. We never meant to imply that transcription cannot or does not affect RT. In fact, we have a recently submitted and have now cited a manuscript (Vouzas et. al.) demonstrating that, when isolated as the only variable, transcription is sufficient to advance RT with a long transcribed transcription unit and to an extent directly proportional to the level of transcription. As such, almost certainly the TSS deletions cause an RT delay due to their effect on transcription, and to different extents in different deletion contexts. In fact, it is quite sensible that the transcriptional and non-transcriptional roles of subERCEs cooperate to elicit robust RT changes during differentiation, which we think is what the reviewer is saying.

Other explanations are also possible. Transcription alters the positions of MCM complexes that are known to be cleared from the bodies of active genes. One thing that often gets forgotten is that if equally early firing origins are distributed more broadly in a population of cells, then any given genomic bin will appear to replicate later in the averaged population data even though the origins are firing at the same time in both cases. Thus the absence of transcription across a large gene like *Morc1* could cause an apparent delay in RT that is actually not a delay but, rather, early replication firing at more diverse sites in different cells.

Regardless of the mechanism, there are clearly elements that maintain early replication in the absence of the vast majority of transcription (subERCEs) and others that influence RT through transcription (e.g. BTSS), which is our main point. We hope that we get closer to the reviewers' concern here by defining "subERCEs" as elements that can advance RT through their ability to enhance transcription but that also have non-transcriptional roles in RT. **We have now devoted a paragraph of the new Discussion section to develop these points.**

In summary, this mutant not only shows weak transcription along the entire locus, but also a small shift towards later replication. Therefore, the authors' conclusion "that transcription is not necessary for early RT at the *Dppa* domain" is not convincingly demonstrated.

Replication is still early after the deletions of BTSS or all three of the TSSs but they eliminate the vast majority of transcription from this locus. **We have re-worded, throughout the manuscript, statements to the effect that "transcription is not necessary for early replication" to statements to the effect that "most transcription is not necessary for early replication."** **We have also supplied a supplemental figure with every replicate of BrU-seq that we have done.**

8) Of the three independent TSS deletions, only the B gives a modest RT delay. So the authors have made a deletion in a context where ERCE A and C are already removed (Δ ABTSSC) and this further increases the RT delay, but less than the full Δ ABC. The authors are trying to make a suggestion that is not entirely satisfactory.

We are not sure what that suggestion is, but we did not mean to imply that transcription plays no role, as addressed in the response to criticism #7 above.

For all these reasons, the demonstration that "transcription is not necessary for early RT of the *Dppa* Domain" is not well supported. How strong sites of replication initiation are involved is

also not well described in this form of the paper. It seems more plausible that a complex cooperation between all these elements is important for the efficient establishment and maintenance of the domain. This model has been proposed in several studies attempting to address this specific question, and I think that the paper would be greatly improved if the authors put previous data into perspective.

We have no data that speaks to establishment. It would have been helpful for the reviewer to cite these “several studies” so we could understand the criticism more clearly. Hopefully the point that most transcription is not necessary for early RT is now clear from above discussion. It is clear that both transcription and subERCs (and “something else” we are still missing) cooperate to give the final RT of this locus. It is also clear that most transcription can be eliminated and replication is still early. In defining subERCs, we are saying that, in addition to transcription, there is another, important, non-transcriptional activity that can advance RT. **We now provide a paragraph in the Discussion of how transcription and subERCs could cooperate to elicit (hypothesis) and maintain (data in this ms) early RT at this domain.**

Minor comments

1) Figure 1 shows only the data already published in Sima et al, 2019. Although it is important to have this figure as a reference (note that ΔC , ΔAC and ΔBC are also shown in Figures 2, 3 and 4 to facilitate the comparison of the profiles), it could be included as a supplementary figure and only discussed mainly in the introduction.

We have moved Figure 1 to supplemental.

2) As mentioned above, there is no visible difference in RT within the Dppa domain when a large deletion of the left part of ERCE C ($\Delta C_{Left20k}$) is made. However, one can observe a difference in the RT of a small domain located at position 50 on Chr16. Surprisingly, the RT of the modified allele is more advanced (Figure 2B). This is also the case for several mutants ($\Delta COSN123$, figure 2B...). Can the authors give us an interpretation of these results?

We have consistently observed this phenomenon in our current and published deletion series. This domain has been shown to display a genetic difference in RT between *m. castaneus* and *m. musculus* as well as high cell to cell variability in RT. To further address this, we correlated the changes in this domain to the changes in the Dppa domain and found no such correlation.

We have now added the following sentence to the end of the legend of Supplementary Fig. 3: “The replication domain to the right of Dppa (c16: 49.65-50.05Mb mm10) shows some clonal variation in RT but there is a poor correlation ($R = -0.35$) between the effects of the deletions in the Dppa domain to this variation. Rather, this domain has been shown to display a genetic difference in RT between *m. castaneus* and *m. musculus* (Rivera-Mulia et al., 2018) as well as high cell to cell variability in RT (Zhao et al., 2020).”

3) The resolution and quality of the figures are really poor and makes it even more difficult to follow the story.

All figures have been completely revised. We hope that the reviewer is satisfied with the presentation of the new figures.

Study of ERCE A and B

4) This part starts with deletions within the ERCE A. To make it easier to understand, I would put the RT profiles being compared on the same line and in the order in which they appear in the text. So on a first line I would put ΔAC (Not really necessary); $\Delta AOSN1COSN123$ and $\Delta ACOSN123$ together.

We have done this in the revised manuscript along with sub-labels for each graph..

5) There is only one clone described for $\Delta BOSN12COSN123$.

As discussed above in regards to $\Delta CRight11.8k$, we only could obtain one clone for several deletion lines. We think it is worth showing the data and we have been transparent regarding which deletions have only single clones. These deletion cell lines take a lot of work to construct and genotype, and it is remarkable that we have independent clones for the majority of our deletions. Almost all deletion studies in the literature use one single clone or animal.

Referee #2:

The manuscript "Master transcription factor binding sites are necessary for early replication control element activity" by Turner et al detailed the cis elements, early replication control elements (ERCEs), that is necessary for early replication in mESCs. They Used CRISPR-Cas9 to map the sites of 3 ERCEs within the Dppa domain. They deleted putative cis elements on one allele while the other allele can serve an internal control in mESCs derived from hybrid mice (F121-9). They found that one or more sites for co-binding of the pluripotency transcription factors Oct4, Sox2, Nanog (OSN) are required for early replication, whereas TSSs are not. So they propose that these master pluripotent transcription factor binding sites (OSN sites) are important for early replication through a transcription-independent way. Overall, the study is well-designed and represents a significant progress from their 2019 original work. With this said, I have the following comments.

Major concerns

1. OSN sites mapped here might still be too large to exclude elements other than OSN sites.

This is certainly true. In fact we know even from the original study that OSN is not sufficient to advance RT because OSN sites Y and Z are still present in the ABC deletion yet RT is completely late. Thus, the activity is unlikely to be OSN per sé. We may have mistakenly over-emphasized the importance of Oct4, Sox2 and Nanog in naming these OSN sites; it was meant to be a descriptive term for the sites we targeted for deletion. **To address this concern, we have now transitioned during the manuscript from OSNs to subERCEs at a point in the results where the activities manifest themselves. To illustrate how complex these sites are in terms of TF binding, we have now performed a motif analysis of the deleted regions and we have mined public ChIP datasets in the Remap 2022 database (Hammal F. et al., NAR, 2022) for TF binding to these sites in mouse ESCs. We present that data in new Supplemental Figure 5, and discuss it in detail in the new Discussion section.**

For example, are there FoxP binding sites in OSN sites?

We are currently not aware of any FoxP1 ChIPseq datasets in mouse ESCs, and none were found in Remap. **We present data from FoxP3 in the new Supplemental Figure 5.** Regardless, we are unclear how this knowledge will get us closer to mechanism. We are not sure of what the exact Fkh1/2 homologues are in mouse and whether mice use the same TFs

to regulate RT. There are myriad factors binding to the subERCEs. The main point of the new figure underscores the reviewers point: there are lots of TFs bound to these sites.

One of the doable convincing strategies is to artificially recruit OSN TFs, or one of them or DBF4 as discussed in the next points, in the cis-deleted background like Δ ABCOSN123, to see whether OSN sites are passable or not. If it works, the DNA-binding mutants of these TFs can be used as an ideal control. All these will also amend the complete missing of the trans factor part in current stage.

The reviewer is asking for a lot here. We have tried artificial recruitment of Gal4-Oct4 fusion to a Gal4 binding site and did not see a shift in RT. Recruiting all three to an ectopic site is a lot of work with technical caveats. The fusion protein itself may be compromised. More importantly, as I mentioned above, our data show that OSN are not sufficient to advance RT even at the native locus. **To illustrate how difficult this experiment would be for readers, we have highlighted the non-sufficiency of OSN in the new version of the manuscript and defined “subERCEs” as sites of diverse TF binding. We also state in the Discussion section that we do not have enough data to investigate a particular TF at this stage of the research, but that is clearly a major goal for the future. In fact it very well may be that the activity is a combination of many TFs, or even different combinations of TFs, as we now state in the Discussion**

We do not agree that DDK recruitment would say anything about the role of TFs or ERCEs. If targeted in an active form, it would be expected to advance replication timing through direct initiation mechanisms downstream of ERCEs, similar to several well defined mechanisms advancing RT in yeasts.

2. Since DDK defines the first kinase to activate MCM helicase during S phase. Have authors checked their enrichment by DDK-ChIP? This may also help to address why OSN (Z) upstream AOSN, X and Y have no ERCE activity.

We are not aware of any DDK ChIP datasets. It, along with many replication initiation factors, have been notoriously difficult to map for unknown reasons. Very few replication proteins have been successfully ChIPed, although many labs have tried. The few reported successes are not with DDK and not in mESCs unfortunately. **To try to address the reviewer's question, we now show in new Supplemental Figure 5 a motif and Remap ChIP analysis comparing all OSNs including X and Y. We also provide a new Figure 5 showing all the origin mapping data available in the literature and don't see subERCEs aligning with efficient origins.**

3. The authors deleted the whole set of all three OSN sites, and conclude a co-binding model. Did authors try to further delete all binding sites for one of these TFs within ERCEs?

This would be impossible. First, the motifs for Oct4 and Sox2 are nearly overlapping and are only consensus sequences. Nanog's motif is unvalidated. Second, it would be impossible to design Cas9 gRNAs to be so surgical. Finally, not all in vivo binding sites have a consensus sequence so deleting a motif does not ensure lack of binding. **We hope that the new Supplemental Figure 5 will clarify some of these questions by showing the sequence and epigenetic anatomy of all OSN sites and the number of motifs and proteins bound just in these few sites. We also discuss limitations of our Crispr deletion approach to make multiple deletions in a single allele and allude to the fact that novel methods may be able to address these complexities, citing a 2023 paper identifying facilitator**

elements in super-enhancers using a complicated approach with synthetic constructs cloned into BACs as an example (Blayney et. al.).

4. Z represents an OSN site near A that, like X and Y shown in Fig.1, does not harbor ERCE activity

Correct. In fact, as pointed out above, this is a very good reason NOT to do the OSN targeting experiments suggested in major concern #1. Clearly OSN binding is not sufficient on its own. **We now make this clear in the manuscript in the new Discussion section and in new Supplemental Figure 5. We also discuss features that may distinguish subERCEs from the Y and Z elements; for example, we show recent high resolution HiChIP and microC data that suggest that Y and Z only weakly interact with ERCEs.**

5. In Discussion part, authors proposed that OSN TFs or other OSN-interacting TFs might be required. Did they try to deplete them or one of them (as in the co-binding model) to test it?

We did deplete Oct4, Sox2 and Nanog alone and in combinations and we did get delays in RT correlated with the locations of OSN sites. However, interpretation of these data is complicated by the fact that OSN regulates pluripotency and many domains are shifting due to spontaneous differentiation. More importantly, Y and Z are OSNs and clearly not sufficient for early replication. Given the number of TFs bound at these sites, we cannot know which one to deplete. Depleting them all in all combinations would be quite a project!! **We now include a proper Discussion section that discusses the reviewers' points.**

6. In Discussion part, an alternative possibility is that OSN sites directly mediate 3D chromatin interactions, which also explains transcription-independent, or even OSN TFs-independent. Agreed -we thought we made it clear that this is part of our model, although we do not think it is OSN as discussed above. **We now provide a Figure 6 model that hopefully addresses this point and a Supplementary Fig. 6 that addresses 3D interactions with recent Hi-ChIP and microC data.**

Minor comments:

1. It would be better to describe how RT is analyzed statistically in details to conclude a significant change/delay or not, for example ΔC , $\Delta C_{Left20k}$ and $\Delta C_{Right14.6k}$ in Fig 2B.

The reviewer makes an important point similar to Reviewer #1 that we did not do a proper statistical analysis. To address this, we developed a means to quantify statistical significance for each of our mutant cell lines. **We present these statistics in each RT panel of the manuscript and we supply a supplemental Figure 2 describing our statistical test and we provide Supplemental Table 4 with the significance of differences between key pairwise comparisons.**

2. The last sentence of Abstract, "pluripotency transcription factor binding sites ensure early replication independent of transcription, suggesting a means for co-regulation of RT with cell fate transitions during development." I cannot clearly understand the logic connection between "independent of transcription" and co-regulation. Pls clarify it in Discussion.

We meant to convey that by having transcription factors regulate RT independent of transcription, RT changes can be elicited by TRN changes during cell fate transitions. **We have now reworded this phrase to read: "Our results suggest a model in which subERCEs respond to diverse master transcription factors by functioning both as transcription enhancers and as elements that organize chromatin domains structurally"**

and support early RT, potentially providing a feed-forward loop to drive robust epigenomic change during cell fate transitions.”

Referee #3:

The authors conducted deletion analyses of the ERCEs A, B and C, with a particular focus on Oct4, Sox2, Nanog (OSN) binding sites. The two major conclusions from this study are 1) OSN binding sites present in these sequences are required for early replication control. 2) Early replication does not require active transcription. The replication timing data presented are convincing and are consistent with the conclusions.

1 Unfortunately, the study fails to provide precise mechanistic insight into the early replication control by the OSN binding. Although several possibilities can be envisioned, the experiments to test the models have not been performed.

With novel or complex biological phenomena, mechanism arises through asking questions at the level that they present themselves. For example, elucidation of enhancers was done by deletion studies near genes in *Drosophila* to find discrete cis-acting elements regulating transcription in a tissue-specific manner. The very recent discovery of facilitators (Blayney et. al.) showed that they function somehow to enhance the activity of enhancers without actually being enhancers, but they did not show any mechanism. Here, we have reduced the size of ERCEs by over 10 fold, and identified a novel activity that can advance RT independent of transcription, that we refer to as subERCEs, to distinguish them from elements that advance RT only through transcription. **To assist readers in appreciating the main messages here, we have provided a model Figure 6 to illustrate our findings of separate activities within the previously defined ERCEs.**

For examples, authors eluded the potential similarity between the actions of pluripotent transcription factors and budding yeast Fkh1/Fkh2. This could be a good analogy, which can be experimentally tested.

We have now made it clear that OSN is a descriptive term for the sites that we identified but that OSN cannot be sufficient for early RT. Rather the elements that we now call subERCEs consist of binding sites for diverse master TFs. Since we do not know which TF, or whether it is a complex combination of TFs, we cannot have knowledge of specific residues with particular functions in any of the TFs, as was the case for mutation of the dimerization domain of yeast Fkh1/2. It would be impossible at this time to carry out analogous experimental manipulations akin to those performed in budding yeast. Even if we did have such information, it would be a tremendous amount of work to carry out such an experiment, well beyond the scope of this work. We agree these are important questions, but respectfully argue that we already present a large body of work and that the question the reviewer wishes to address would constitute a PhD thesis in itself. **We do, however, suggest this and other ideas as future directions in our Discussion.**

2 Obvious questions arising from the data presented in the manuscript have not been experimentally addressed.

2-1 What is the epigenetic state of the OSN mutants? Related to this question, what is the correlation between the superenhancer and what the authors call "replication enhancer"?

Performing correlative analysis of epigenetic marks in multiple mutant cell lines would be a tremendous fishing expedition that certainly would not get us to the mechanisms that the reviewer would like.

The analogy to superenhancers is a very good one. As we stated in 2019, there is good but not perfect overlap of ERCES with superenhancers (both defined across large genomic regions). Superenhancers are also compound elements consisting of enhancers (superenhancers to enhancers / ERCES to subERCES). Indeed, subERCES may also be enhancers within subenhancers.. **We now provide a Supplementary Figure 6 with a model drawing to illustrate what we know and we discuss analogies to super-enhancers in the Discussion.**

2-2 What is the chromatin openness of the mutant cell lines? Does open chromatin induced by the OSN binding explain the early replication?

TSSs, subERCES and X and Z are DNaseI / ATAC-seq sensitive sites, so this does not distinguish elements that regulate early replication. **Figures 1-3 and Supplemental Fig. 1 show the anatomy of ERCES with respect to chromatin marks and ATAC-seq.**

3 Data show that deletion of TSS abrogates transcription but does not significantly affect early replication, showing that the transcription is not required for early replication. How about the transcription in these OSN binding site mutants? Is it affected by the OSN binding sites mutant cells[;]x/]?

In revised Figure 4, we now show BrU-seq in several new mutants, including Δ ABOSN12COSN123 with and without the additional deletion of BTSS. OSN deletions do impact transcription, likely because they are enhancers. We discuss the dual role of subERCES in RT and transcription in the new Discussion section.

4 Other questions:

4-1 This study shows that OSN binding sites are required for early replication, but it is not clear if they are sufficient. Can authors reconstruct the early replication by combining the OSN binding sequences?

Insertion of these sites into ectopic sites is the project of a new post-doc and is an entire study in itself. **We now mention in the Discussion that deletions have their limitations and that we need new approaches, such as insertions.**

4-2 X,Y and Z represent the OSN that are not required for ERCE. What are the differences between these and those OSN binding sites in ERCE. Can the former replace the latter? Or are there intrinsic differences between these OSN binding sites?

New Supplementary Figure 5 compares X, Y and Z to the ERCE-resident OSN sites for their composition. This figure also identifies differences in interaction frequency of these elements with ERCES vs. subERCES. Swapping experiments would be a big project and the exact context is likely important. This requires a devoted effort; for example, in performing ectopic insertions, we will also insert X, Y and Z.

General suggestion on the nomenclature of the mutant cell lines:

I think it would be easier for the readers to understand each deletion cell lines if nomenclatures are changed a bit.

All the cell lines are combinations of different deletions. Each deletion could be preceded by Δ . (e.g. Δ AOSN1 Δ COSN123 rather than Δ AOSN1COSN123; Δ A Δ B Δ COSN123 rather than Δ ABCOSN123). This would make it easier for the readers to grasp the combination of the mutations from each domain.

This was a very helpful suggestion, we have modified the nomenclature by using only one Δ symbol and adding a dot in between each element deleted. We hope that the reviewer likes this new nomenclature.

In Figure 3C

Δ AOSNBCOSN123 should be Δ AOSN1BCOSN123 (Δ AOSN1 Δ B Δ COSN123)

ZAOSNCOSN123 should be Δ ZAOSN1COSN123 (Δ Z Δ AOSN1 Δ COSN123)

ZBC should be Δ ZBC (Δ Z Δ B Δ C)

DONE

Figure 1 is the summary of the previous publications, and can be presented as a Supplementary Figure.

DONE

Prof. David M Gilbert
San Diego Biomedical Research Institute

26th Mar 2025

Re: EMBOJ-2023-116114R-Q
Master transcription factor binding sites constitute the core of early replication control elements

Dear Dr. Gilbert,

Thank you for resubmitting a new version of your manuscript for our consideration. I sent it once more to the two referees with the most pertinent concerns during the original review, and they have in the meantime returned the reports copied below. I am happy to say that both of them consider the study significantly improved and would in principle support its publication in The EMBO Journal now. Nevertheless, referee 3 still retains a number of specific concerns and queries, which I would ask you to respond to during a final, formal round of revision now. This would not necessarily require additional experimentation, but any data that you may already have to answer the open questions should be very helpful for further strengthening the work.

For this formal round of revision, please also note and carefully address a number of editorial issues at this stage, as follows:

- Please adjust the order of the manuscript sections: Title page with complete author information, Abstract, Keywords, Introduction, Results, Discussion, Methods, Data Availability, Acknowledgements, Disclosure and Competing Interests Statement, References, Main Figure Legends, Tables, Expanded Figure Legends.
- Please include a dedicated "Data Availability" section at the end of the Material and Methods (suggested wording: "The [structural coordinates | microarray | mass spectrometry] data from this publication have been deposited to the [name of the database] database [URL] and assigned the identifier [accession | permalink | hashtag]."); should there no data deposition to public repositories linked to the study, this should still be stated as "This study includes no data deposited in external repositories."
- Please indicate the statistical test used for data analysis in the legends of figures 1D, 2D, 3D, 4D, "supplementary" figures 1D, 2D.
- Please convert the "supplemental figures" into "expanded view" figures, renaming them in the legends and in-text call-outs to "Figure EV1-5". Since "Supplemental Figure S3" does only have one main panel, please remove the letter "A" from the figure. Please upload each main and each EV figure as individual files in a figure (rather than PPTX) format, with sufficient resolution/quality for production.
- Please rename the five "supplemental" tables into "expanded view tables", renaming them in the files and in-text call-outs to "Table EV1-5". Their legends should be removed from the main text and instead be included in a separate "legends" tab in the XLSX files for Tables EV1, 2, 4, or above the table in (editable) Word/Excel/etc files for Tables EV3 and EV5.
- On the abstract page of the manuscript, please include 4-5 general keyword terms to enhance searchability.
- Please carefully check the bibliography for completeness (journal, year, volume, pagination...) of all references. Also, please adjust the format for preprint citation as specified in our author guidelines. The citation in the text should be: "(PREPRINT: name1 et al, year)"; in the reference list: "Author name1, Author name2, ... (year) article title. bioRxiv doi: XXX"
- Please remove the Reagents and Tools table from the main manuscript file and instead upload it as a separate file, using the dedicate template file that can be downloaded from our Guide to Authors (see also information & links below).
- As we are switching from a free-text author contribution statement towards a more formal statement based on Contributor Role Taxonomy (CRediT) terms, please remove the present Author Contribution section and instead specify each author's contribution(s) directly in the Author Information page of our submission system during upload of the final manuscript. See <https://casrai.org/credit/> for more information.
- Please rename the Competing Interest section into "Disclosure and Competing Interests Statement", in accordance with our updated Guide to Authors (<https://www.embopress.org/competing-interests>)
- Please provide suggestions for a short 'blurb' text prefacing and summing up the study in two sentences (max. 250 characters), followed by 3-5 one-sentence 'bullet points' with brief factual statements of key results of the paper; they will form the basis of an editor-written 'Synopsis' accompanying the online version of the article. Please also upload a synopsis image, which can be

used as a "visual title" for the synopsis section of your paper (maybe based on a simplified version of Fig 6AB?). The image should be in PNG or JPG format with the modest dimensions of EXACTLY 550 pixels wide and 300-600 pixels high.

- Finally, you shall also receive a separate message from our Source Data curation team, with instructions on how to prepare and upload relevant image and numerical raw data.

Please do not hesitate to contact me should you have any questions regarding the remaining scientific revisions, or any of the editorial points! I look forward to receiving your final version.

Yours sincerely,

Hartmut Vodermaier

- 1) Every manuscript requires a Data Availability section (even if only stating that no deposited datasets are included). Primary datasets or computer code produced in the current study have to be deposited in appropriate public repositories prior to resubmission, and reviewer access details provided in case that public access is not yet allowed. Further information: embopress.org/page/journal/14602075/authorguide#dataavailability
- 2) Each figure legend must specify
 - size of the scale bars that are mandatory for all micrograph panels
 - the statistical test used to generate error bars and P-values
 - the type error bars (e.g., S.E.M., S.D.)
 - the number (n) and nature (biological or technical replicate) of independent experiments underlying each data point
 - Figures may not include error bars for experiments with $n < 3$; scatter plots showing individual data points should be used instead.
- 3) Revised manuscript text (including main tables, and figure legends for main and EV figures) has to be submitted as editable text file (e.g., .docx format). We encourage highlighting of changes (e.g., via text color) for the referees' reference.
- 4) Each main and each Expanded View (EV) figure should be uploaded as individual production-quality files (preferably in .eps, .tif, .jpg formats). For suggestions on figure preparation/layout, please refer to our Figure Preparation Guidelines: <http://bit.ly/EMBOPressFigurePreparationGuideline>
- 5) Point-by-point response letters should include the original referee comments in full together with your detailed responses to them (and to specific editor requests if applicable), and also be uploaded as editable (e.g., .docx) text files.
- 6) Please complete our Author Checklist, and make sure that information entered into the checklist is also reflected in the manuscript; the checklist will be available to readers as part of the Review Process File. A download link is found at the top of our Guide to Authors: embopress.org/page/journal/14602075/authorguide
- 7) All authors listed as (co-)corresponding need to deposit, in their respective author profiles in our submission system, a unique ORCID identifier linked to their name. Please see our Guide to Authors for detailed instructions.
- 8) Please note that supplementary information at EMBO Press has been superseded by the 'Expanded View' for inclusion of additional figures, tables, movies or datasets; with up to five EV Figures being typeset and directly accessible in the HTML version of the article. For details and guidance, please refer to: embopress.org/page/journal/14602075/authorguide#expandedview
- 9) To facilitate reproducibility and cross-laboratory adoption of methodologies, please structure the Materials & Methods section as outlined in our guide to authors, including a completed Reagents and Tools Table that can be downloaded from our author guidelines as well (<https://www.embopress.org/page/journal/14602075/authorguide#structuredmethods>).
- 10) Digital image enhancement is acceptable practice, as long as it accurately represents the original data and conforms to community standards. If a figure has been subjected to significant electronic manipulation, this must be clearly noted in the figure

legend and/or the 'Materials and Methods' section. The editors reserve the right to request original versions of figures and the original images that were used to assemble the figure. Finally, we generally encourage uploading of numerical as well as gel/blot image source data; for details see: embopress.org/page/journal/14602075/authorguide#sourcedata

At EMBO Press, we ask authors to provide source data for the main manuscript figures. Our source data coordinator will contact you to discuss which figure panels we would need source data for and will also provide you with helpful tips on how to upload and organize the files.

In the interest of ensuring the conceptual advance provided by the work, we recommend submitting a revision within 3 months (24th Jun 2025). Please discuss the revision progress ahead of this time with the editor if you require more time to complete the revisions. Use the link below to submit your revision:

Link Not Available

Referee #1:

The authors gave very satisfactory answers to all the questions I raised. The article has gained in clarity and impact. As rightly mentioned in the response to the referees, this is a considerable amount of work that provides important information on the complexity of the cis-elements involved in the establishment and maintenance of early replicating domains regulated during differentiation. The new discussion has been considerably improved and places this new work into the context of current knowledge. For all these reasons, I support the publication of this article in EMBO journal.

I have noted a few minor editorial errors.

Introduction third paragraph: the face of of severe.....

In the discussion, paragraph ERCEs consist of multiple....

Second line: Here, we show here...

Referee #3:

In this manuscript, Turner et al. have conducted detailed analyses of the genetic elements required for maintaining early replication timing (RT) of the Dppa2/4 replication domain. The authors have utilized heterozygous mice cell line, that permitted them to conduct precise and quantitative RT analyses of the mutant allele in comparison with the wild-type allele. They also developed a novel method to measure and statistically assess the effect of each deletion on RT, as well as to compare deletions to each other.

They previously identified three DNA segments that are required for early RT of the Dppa2/4 replication domain. In this manuscript, they further dissected each segment, and identified what they call now "subERCE" which may be composed of binding sites of multiple transcription factors. Combinations of the deletion of three subERCEs in the C region with deltaA and deltaB could convert the early RT to late RT to full extent. However, subERCEs in A and B could not reconstitute the entire deletion, suggesting the presence of some other unidentified genetic elements. They have also shown that early RT can be maintained in the absence of most transcription and that early RT is independent of where replication initiates.

The findings in this manuscript provide novel insight into the genetic elements required for maintenance of early RT, and are of general significance and appeal to the wider readership of EMBO Journal, although it is currently not clear how this is related to the early to late conversion during the differentiation of this region, how this mechanism is universal in other early RT domains, and how ERCEs are related to superenhancers.

Major comments:

1 One of the important conclusions of this study is that "Early RT can be maintained in the absence of most transcription". I agree with this conclusion.

Is the loss of Early RT by defect in ERCE function always associated with loss of transcription? In other words, are there any mutants that are transcriptionally active, but is completely deficient in early RT (the entire RT segment rendered to be late-replicating as seen in deltaA.B.C.)?

2 With regard to the potential effect of subERCEs on the open chromatin structure, I understand that they are ATAC sensitive sites. However, the deletion of one subERCE may cause more wide-spread effect. Does deletion of one subERCE affect the chromatin openness at other subERCEs? Authors postulate the chromatin interactions that may contribute to the generation of the DPPA RT domain. These interactions may serve for maintenance of open chromatin structures.

Also, TSSs also overlap with ATAC sensitive sites. Does deletion of a TSS affect the ATAC sensitivity of the neighboring subERCEs?

3 Related to the superenhancer/ ERCE analogy, H3K27as is clustered near the subERCEs/TTS segments. The analyses of H3K27as in selected mutant cell lines would give important information as to the potential mechanism of the subERCE functions.

4 It is quite obvious that deletion of A, B or C causes regional effects on RT. deltaC causes RT conversion at the right half of the domain. delta B causes slight RT change in the central part. delta BC causes larger RT change at the center-to-right portion of the domain, whereas deltaAB causes larger RT change at the center-to-left portion of the domain. delta AC causes larger RT change at both left and right portions of the domain. deltaA does not cause RT change, but in combination with delta B or delta C, it causes larger RT change.

Authors could discuss potential implications of these facts in relation to their model.

The lack of much effect of deltaA alone could be due to the size of the deletion. I wonder if the larger deletion including X and Z would cause RT change on the left half of the domain.

Minor comments:

Discussion "ERCEs consist of multiple smaller "subERCEs".

11th line from the bottom

Supplemental Figure B

should be

Supplemental Figure 5B

Discussion last page

while Y and Z interact do not interact with subERCEs.

->

while Y and Z do not interact with subERCEs.

David M. Gilbert, Ph.D.
Professor
3525 John Hopkins Court
San Diego, CA 92121
Email: gilbert@sdbri.org

June 15, 2025

RE: EMBOJ-2023-116114R-Q

Dear Dr. Vodermaier,

Please find our revised manuscript “**Master transcription factor binding sites constitute the core of early replication control elements**”.

We want to apologize again for taking over two months to respond. This manuscript has quite a lot of datasets to wrangle and GEO had a lot of technical questions.

On the following pages, we provide a point by point response to the final requests, starting with your editorial requests and following with the reviewers final concerns. As before, our responses are *in blue font*, with changes made to the manuscript *in bold blue font*:

Sincerely,

David M. Gilbert, Ph.D.
Professor
San Diego Biomedical Research Institute

EDITORIAL REQUESTS

- Please adjust the order of the manuscript sections: Title page with complete author information, Abstract, Keywords, Introduction, Results, Discussion, Methods, Data Availability, Acknowledgements, Disclosure and Competing Interests Statement, References, Main Figure Legends, Tables, Expanded Figure Legends.

DONE

- Please include a dedicated "Data Availability" section at the end of the Material and Methods (suggested wording: "The [structural coordinates | microarray | mass spectrometry] data from this publication have been deposited to the [name of the database] database [URL] and assigned the identifier [accession | permalink | hashtag]."); should there no data deposition to public repositories linked to the study, this should still be stated as "This study includes no data deposited in external repositories."

DONE

- Please indicate the statistical test used for data analysis in the legends of figures 1D, 2D, 3D, 4D, "supplementary" figures 1D, 2D.

DONE

- Please convert the "supplemental figures" into "expanded view" figures, renaming them in the legends and in-text call-outs to "Figure EV1-5". Since "Supplemental Figure S3" does only have one main panel, please remove the letter "A" from the figure. Please upload each main and each EV figure as individual files in a figure (rather than PPTX) format, with sufficient resolution/quality for production.

DONE

- Please rename the five "supplemental" tables into "expanded view tables", renaming them in the files and in-text call-outs to "Table EV1-5". Their legends should be removed from the main text and instead be included in a separate "legends" tab in the XLSX files for Tables EV1, 2, 4, or above the table in (editable) Word/Excel/etc files for Tables EV3 and EV5.

DONE

- On the abstract page of the manuscript, please include 4-5 general keyword terms to enhance searchability.

DONE

- Please carefully check the bibliography for completeness (journal, year, volume, pagination...) of all references. Also, please adjust the format for preprint citation as specified in our author guidelines. The citation in the text should be: "(PREPRINT: name1 et al, year)"; in the reference list: "Author name1, Author name2, ... (year) article title. bioRxiv doi: XXX"

DONE

- Please remove the Reagents and Tools table from the main manuscript file and instead upload it as a separate file, using the dedicate template file that can be downloaded from our Guide to Authors (see also information & links below).

DONE

- As we are switching from a free-text author contribution statement towards a more formal statement based on Contributor Role Taxonomy (CRediT) terms, please remove the present Author Contribution section and instead specify each author's contribution(s) directly in the Author Information page of our submission system during upload of the final manuscript. See <https://casrai.org/credit/> for more information.

DONE

- Please rename the Competing Interest section into "Disclosure and Competing Interests Statement", in accordance with our updated Guide to Authors (<https://www.embopress.org/competing-interests>)

DONE

- Please provide suggestions for a short 'blurb' text prefacing and summing up the study in two sentences (max. 250 characters), followed by 3-5 one-sentence 'bullet points' with brief factual statements of key results of the paper; they will form the basis of an editor-written 'Synopsis' accompanying the online version of the article. Please also upload a synopsis image, which can be used as a "visual title" for the synopsis section of your paper (maybe based on a simplified version of Fig 6AB?). The image should be in PNG or JPG format with the modest dimensions of EXACTLY 550 pixels wide and 300-600 pixels high.

These are provided in two separate files named:

- Blurb and Bullet Points for EMBOJ-final
- EMBO_visualAbstract_v2

REQUESTS FROM SOURCE DATA CURATION TEAM

Source data and source data checklist have been submitted

FORMATTING GUIDELINE REQUESTS

DONE

DONE

DONE

DONE

DONE

DONE

DONE

DONE

9) To facilitate reproducibility and cross-laboratory adoption of methodologies, please structure the Materials & Methods section as outlined in our guide to authors, including a completed Reagents and Tools Table that can be downloaded from our author guidelines as well (<https://www.embopress.org/page/journal/14602075/authorguide#structuredmethods>).

DONE

10) Digital image enhancement is acceptable practice, as long as it accurately represents the original data and conforms to community standards. If a figure has been subjected to significant electronic manipulation, this must be clearly noted in the figure legend and/or the 'Materials and Methods' section. The editors reserve the right to request original versions of figures and the original images that were used to assemble the figure. Finally, we generally encourage uploading of numerical as well as gel/blot image source data; for details see: embopress.org/page/journal/14602075/authorguide#sourcedata

DONE

At EMBO Press, we ask authors to provide source data for the main manuscript figures. Our source data coordinator will contact you to discuss which figure panels we would need source data for and will also provide you with helpful tips on how to upload and organize the files.

DONE

RESPONSES TO REVIEWERS

Referee #1:

The authors gave very satisfactory answers to all the questions I raised. The article has gained in clarity and impact. As rightly mentioned in the response to the referees, this is a considerable amount of work that provides important information on the complexity of the cis-elements involved in the establishment and maintenance of early replicating domains regulated during differentiation. The new discussion has been considerably improved and places this new work into the context of current knowledge. For all these reasons, I support the publication of this article in EMBO journal.

I have noted a few minor editorial errors.

Introduction third paragraph: the face of of severe.....

In the discussion, paragraph ERCEs consist of multiple....

Second line: Here, we show here...

Both done. Thank you!!

Referee #3:

In this manuscript, Turner et al. have conducted detailed analyses of the genetic elements required for maintaining early replication timing (RT) of the Dppa2/4 replication domain. The authors have utilized heterozygous mice cell line, that permitted them to conduct precise and quantitative RT analyses of the mutant allele in comparison with the wild-type allele. They also developed a novel method to measure and statistically assess the effect of each deletion on RT, as well as to compare deletions to each other.

They previously identified three DNA segments that are required for early RT of the Dppa2/4 replication domain. In this manuscript, they further dissected each segment, and identified what they call now "subERCE" which may be composed of binding sites of multiple transcription factors. Combinations of the deletion of three subERCEs in the C region with deltaA and deltaB could convert the early RT to late RT to full extent. However, subERCEs in A and B could not reconstitute the entire deletion, suggesting the presence of some other unidentified genetic elements. They have also shown that early RT can be maintained in the absence of most transcription and that early RT is independent of where replication initiates.

The findings in this manuscript provide novel insight into the genetic elements required for maintenance of early RT, and are of general significance and appeal to the wider readership of EMBO Journal, although it is currently not clear how this is related to the early to late conversion during the differentiation of this region, how this mechanism is universal in other early RT domains, and how ERCEs are related to superenhancers.

With respect to RT switches during differentiation, in the prior Discussion, we discussed how changes in RT would be related to cell fate transitions. **To make this more clear, we now explicitly say in the final paragraph of the Discussion: "In this model, the switch to late replication of the Dppa domain upon differentiation would accompany the downregulation of the pluripotency-specific TFs that bind to subERCEs."**

As for universality, we have added a sentence also in the last paragraph of the Discussion as follows: "Given that we have validated ERCEs in two other domains (SIMA 2019), it is likely that the mechanisms functioning at the Dppa domain also function in other developmentally regulated replication domains."

As for ERCEs and superenhancers, we have said all that we feel comfortable saying in the sentence already provided in the Discussion: "It should also be noted that a recent study demonstrated that superenhancers contain bioinformatically equivalent but functionally distinct "facilitators" that potentiate enhancer activity dependent upon their positions within the superenhancer (Blayney *et al*, 2023), raising intriguing parallels to subERCEs within ERCEs." Anything more would be pure speculation that can be left up to the imagination of the reader in our opinion.

Major comments:

1 One of the important conclusions of this study is that "Early RT can be maintained in the absence of most transcription". I agree with this conclusion

Is the loss of Early RT by defect in ERCE function always associated with loss of transcription? In other words, are there any mutants that are transcriptionally active, but is completely deficient in early RT (the entire RT segment rendered to be late-replicating as seen in deltaA.B.C.)?

Not substantial levels of transcription. **To clarify this point, in the transcription section of the Discussion, we have added the following sentence: "Deletion of subERCEs severely diminishes transcription, suggesting they do play roles as transcriptional enhancers, and we have no mutants that are late replicating while retaining substantial levels of transcription."**

2 With regard to the potential effect of subERCEs on the open chromatin structure, I understand that they are ATAC sensitive sites. However, the deletion of one subERCE may cause more wide-spread effect. Does deletion of one subERCE affect the chromatin openness at other subERCEs? Authors postulate the chromatin interactions that may contribute to the generation of the DPPA RT domain. These interactions may serve for maintenance of open chromatin structures.

Also, TSSs also overlap with ATAC sensitive sites. Does deletion of a TSS affect the ATAC sensitivity of the neighboring subERCEs?

As we understand it, the reviewer would like us to perform ATAC-seq in a selection of mutants with single subERCE or TSS deletions to determine whether or not the ATAC-seq sensitivity of other subERCEs or TSSs is affected by the loss of one. We are not clear on what hypothesis the reviewer would like to test. It would seem that this experiment would not be to shore up principles presented in this manuscript but rather could lead to interesting preliminary descriptive work for future exploration. Whether or not ATAC-seq sensitivity is changed

elsewhere it would still require a lot of downstream work to understand the mechanism by which RT is altered.

3 Related to the superenhancer/ ERCE analogy, H3K27as is clustered near the subERCEs/TTS segments. The analyses of H3K27as in selected mutant cell lines would give important information as to the potential mechanism of the subERCE functions.

The same caveats apply to this descriptive experiment as to the ATAC-seq descriptive experiment above. Without knowing the reviewer's hypothesis, it is not clear how this experiment could, on its own without substantial follow up, provide mechanistic information.

4 It is quite obvious that deletion of A, B or C causes regional effects on RT. deltaC causes RT conversion at the right half of the domain. delta B causes slight RT change in the central part. delta BC causes larger RT change at the center-to-right portion of the domain, whereas deltaAB causes larger RT change at the center-to-left portion of the domain. delta AC causes larger RT change at both left and right portions of the domain. deltaA does not cause RT change, but in combination with delta B or delta C, it causes larger RT change.

Authors could discuss potential implications of these facts in relation to their model.

We have now added the following sentence to the last paragraph of the Discussion: "Formation of a microenvironment also can explain why ERCE deletions tend to result in shifts of peak early replication to the general region of remaining ERCEs that may promote more localized micro-environments. "

The lack of much effect of deltaA alone could be due to the size of the deletion. I wonder if the larger deletion including X and Z would cause RT change on the left half of the domain.

In our 2019 publication, we showed a 100kb deletion that takes out X, A and Z with a very small but significant effect on RT. This was shown in Supplemental Figure 3 (now called Extended Data 3).

Minor comments:

Discussion "ERCEs consist of multiple smaller "subERCEs". 11th line from the bottom Supplemental Figure B should be Supplemental Figure 5B

Discussion last page

while Y and Z interact do not interact with subERCEs. -> while Y and Z do not interact with subERCEs.

Both done. Thank you!!

Prof. David M Gilbert
San Diego Biomedical Research Institute
Chromosome Replication and Epigenome Regulation

23rd Jun 2025

Re: EMBOJ-2023-116114R1
Master transcription-factor binding sites constitute the core of early replication control elements

Dear Dr. Gilbert,

Thank you for submitting your final revised manuscript for our consideration. I am pleased to inform you that we have now accepted it for publication in The EMBO Journal.

Yours sincerely,

Hartmut Vodermaier
